# THEORETICAL BOUNDS ON ESTIMATION ERROR FOR META-LEARNING

**James Lucas, Mengye Ren, Irene Raissa KAMENI KAMENI, Toniann Pitassi & Richard Zemel**

## ABSTRACT

Machine learning models have traditionally been developed under the assumption that the training and test distributions match exactly. However, recent success in few-shot learning and related problems are encouraging signs that these models can be adapted to more realistic settings where train and test distributions differ. Unfortunately, there is severely limited theoretical support for these algorithms and little is known about the difficulty of these problems. In this work, we provide novel information-theoretic lower-bounds on minimax rates of convergence for algorithms that are trained on data from multiple sources and tested on novel data. Our bounds depend intuitively on the information shared between sources of data, and characterize the difficulty of learning in this setting for arbitrary algorithms. We demonstrate these bounds on a hierarchical Bayesian model of meta-learning, computing both upper and lower bounds on parameter estimation via maximum-a-posteriori inference.

## 1 INTRODUCTION

Many practical machine learning applications deal with distributional shift from training to testing. One example is few-shot classification (Ravi & Larochelle, 2016; Vinyals et al., 2016), where new classes need to be learned at test time based on only a few examples for each novel class. Recently, few-shot classification has seen increased success; however, theoretical properties of this problem remain poorly understood.

In this paper we analyze the *meta-learning* setting, where the learner is given access to samples from a set of meta-training distributions, or tasks. At test-time, the learner is exposed to only a small number of samples from some novel task. The meta-learner aims to uncover a useful inductive bias from the original samples, which allows them to learn a new task more efficiently.[1] While some progress has been made towards understanding the generalization performance of specific meta-learning algorithms (Amit & Meir, 2017; Khodak et al., 2019; Bullins et al., 2019; Denevi et al., 2019; Cao et al., 2019), little is known about the difficulty of the meta-learning problem in general. Existing work has studied generalization upper-bounds for novel data distributions (Ben-David et al., 2010; Amit & Meir, 2017), yet to our knowledge, the inherent difficulty of these tasks relative to the *i.i.d* case has not been characterized.

In this work, we derive novel bounds for meta learners. We first present a general information theoretic lower bound, Theorem 1, that we use to derive bounds in particular settings. Using this result, we derive lower bounds in terms of the number of training tasks, data per training task, and data available in a novel target task. Additionally, we provide a specialized analysis for the case where the space of learning tasks is only partially observed, proving that infinite training tasks or data per training task are insufficient to achieve zero minimax risk (Corollary 2).

We then derive upper and lower bounds for a particular meta-learning setting. In recent work, Grant et al. (2018) recast the popular meta-learning algorithm MAML (Finn et al., 2017) in terms of inference in a Bayesian hierarchical model. Following this, we provide a theoretical analysis of a hierarchical Bayesian model for meta-linear-regression. We compute sample complexity bounds for posterior inference under Empirical Bayes (Robbins, 1956) in this model and compare them to our predicted lower-bounds in the minimax framework. Furthermore, through asymptotic analysis of the error rate of the MAP estimator, we identify crucial features of the meta-learning environment which are necessary for novel task generalization.

---

[1]Note that this definition encompasses few-shot learning.

Our primary contributions can be summarized as follows:

- We introduce novel lower bounds on minimax risk of parameter estimation in meta-learning.
- Through these bounds, we compare the relative utility of samples from meta-training tasks and the novel task and emphasize the importance of the relationship between the tasks.
- We provide novel upper bounds on the error rate for estimation in a hierarchical meta-linear-regression problem, which we verify through an empirical evaluation.

## 2 RELATED WORK

An early version of this work (Lucas et al., 2019) presented a restricted version of Theorem 1. The current version includes significantly more content, including more general lower bounds and corresponding upper bounds in a hierarchical Bayesian model of meta-learning (Section 5).

Baxter (2000) introduced a formulation for inductive bias learning where the learner is embedded in an environment of multiple tasks. The learner must find a hypothesis space which enables good generalization on average tasks within the environment, using finite samples. In our setting, the learner is not explicitly tasked with finding a reduced hypothesis space but instead learns using a general two-stage approach, which matches the standard meta-learning paradigm (Vilalta & Drissi, 2002). In the first stage an inductive bias is extracted from the data, and in the second stage the learner estimates using data from a novel task distribution. Further, we focus on bounding minimax risk of meta learners. Under minimax risk, an optimal learner achieves minimum error on the hardest learning problem in the environment. While average case risk of meta learners is more commonly studied, recent work has turned attention towards the minimax setting (Kpotufe & Martinet, 2018; Hanneke & Kpotufe, 2019; 2020; Mousavi Kalan et al., 2020; Mehta et al., 2012). The worst-case error in meta-learning is particularly important in safety-critical systems, for example in medical diagnosis.

Mousavi Kalan et al. (2020) study the minimax risk of transfer learning. In their setting, the learner is provided with a large amount of data from a single source task and is tasked with generalizing to a target task with a limited amount of data. They assume relatedness between tasks by imposing closeness in parameter-space (whereas in our setting, we assume closeness in distribution via KL divergence). They prove only lower bounds, but notably generalize beyond the linear setting towards single layer neural networks.

There is a large volume of prior work studying upper-bounds on generalization error in multi-task environments (Ben-David & Borbely, 2008; Ben-David et al., 2010; Pentina & Lampert, 2014; Amit & Meir, 2017; Mehta et al., 2012). While the approaches in these works vary, one common factor is the need to characterize task-relatedness. Broadly, these approaches either assume a shared distribution for sampling tasks (Baxter, 2000; Pentina & Lampert, 2014; Amit & Meir, 2017), or a measure of distance between distributions (Ben-David & Borbely, 2008; Ben-David et al., 2010; Mohri & Medina, 2012). Our lower-bounds utilize a weak form of task relatedness, assuming that the environment contains a finite set that is suitably separated in parameter space but close in KL divergence—this set of assumptions also arises often when computing *i.i.d* minimax lower bounds (Loh, 2017).

One practical approach to meta-learning is learning a linear mapping on top of a learned feature space. Prototypical Networks (Snell et al., 2017) effectively learn a discriminative embedding function and performs linear classification on top using the novel task data. Analyzing these approaches is challenging due to metric-learning inspired objectives (that require non-*i.i.d* sampling) and the simultaneous learning of feature mappings and top-level linear functions. Though some progress has been made (Jin et al., 2009; Saunshi et al., 2019; Wang et al., 2019; Du et al., 2020). Maurer (2009), for example, explores linear models fitted over a shared linear feature map in a Hilbert space. Our results can be applied in these settings if a suitable packing of the representation space is defined.

Other approaches to meta-learning aim to parameterize learning algorithms themselves. Traditionally, this has been achieved by hyper-parameter tuning (Rasmussen & Nickisch, 2010; MacKay et al., 2019) but recent fully parameterized optimizers also show promising performance in deep neural network optimization (Andrychowicz et al., 2016), few-shot learning (Ravi & Larochelle, 2016), unsupervised learning (Metz et al., 2019), and reinforcement learning (Duan et al., 2016). Yet another

approach learns the initialization of task-specific parameters, that are further adapted through regular gradient descent. Model-Agnostic Meta-Learning (Finn et al., 2017), or MAML, augments the global parameters with a meta-initialization of the weight parameters. Grant et al. (2018) recast MAML in terms of inference in a Bayesian hierarchical model. In Section 5, we consider learning in a hierarchical environment of linear models and provide both lower and upper bounds on the error of estimating the parameters of a novel linear regression problem.

Lower bounding estimation error is a critical component of understanding learning problems (and algorithms). Accordingly, there is a large body of literature producing such lower bounds (Khas' minskii, 1979; Yang & Barron, 1999; Loh, 2017). We focus on producing lower-bounds for parameter estimation using local packing sets, but expect that extending these results to density estimation or non-parametric estimation is feasible.

## 3 NOVEL TASK ENVIRONMENT RISK

Most existing theoretical work studying out-of-distribution generalization focuses on providing upper-bounds on generalization performance (Ben-David et al., 2010; Pentina & Lampert, 2014; Amit & Meir, 2017). We begin by instead exploring the converse: what is the best performance we can hope to achieve on any given task in the environment? After introducing notation and minimax risks, we then show how these ideas can be applied, using meta linear regression as an example.

A full reference table for notation can be found in Appendix A and a short summary is given here. We consider algorithms that learn in an environment $(\mathcal{Z}, \mathcal{P})$, with data domain $\mathcal{Z} = \mathcal{X} \times \mathcal{Y}$ and $\mathcal{P}$ a space of distributions with support $\mathcal{Z}$. In the typical *i.i.d* setting, the algorithm is provided training data $S \in \mathcal{Z}^k$, consisting of $k$ *i.i.d* samples from $P \in \mathcal{P}$.

In the standard *multi-task* setting, we sample training data from a set of training tasks $\{P_1, \ldots, P_{M+1}\} \subset \mathcal{P}$. We extend this to a meta-learning, or *novel-task* setting by first drawing $S_{1:M}$: $n$ training data points from the first $M$ distributions, for a total of $nM$ samples. We call this the *meta-training set*. We then draw a small sample of novel data, called a *support set*, $S_{M+1} \in \mathcal{Z}^k$, from $P_{M+1}$.

Consider a symmetric loss function $\ell(a, b) = \psi(\rho(a, b))$ for non-decreasing $\psi$ and arbitrary metric $\rho$. We seek to estimate the output of $\theta : \mathcal{P} \to \Omega$, a functional that maps distributions to a metric space $\Omega$. For example, $\theta(P)$ may describe the coefficient vector of a high-dimensional hyperplane when $\mathcal{P}$ is a space of linear models, and $\rho$ may be the Euclidean distance.

**The *i.i.d* minimax risk**  Before studying the meta-learning setting, we first begin with a definition of the *i.i.d* minimax risk that measures the worst-case error of the best possible estimator,

$$R^* = \inf_{\hat{\theta}} \sup_{P \in \mathcal{P}} \mathbb{E}_{S \sim P^k} \left[ \ell(\hat{\theta}(S), \theta_P) \right]. \tag{1}$$

For notational convenience, we denote the output of $\theta(P)$ by $\theta_P$. The estimator for $\theta$ is denoted, $\hat{\theta} : \mathcal{Z}^k \to \Omega$, and maps $k$ samples from $P$ to an estimate of $\theta_P$.

**Novel-task minimax risk**  In the novel-task setting, we wish to estimate $\theta_{P_{M+1}}$, the parameters of the novel task distribution $P_{M+1}$. We consider two-stage estimators for $\theta_{P_{M+1}}$. In the first stage, the meta-learner uses a learning algorithm $f : S_{1:M} \mapsto \hat{\boldsymbol{\theta}}_{S_{1:M}}$, that maps the meta-training set to an estimation algorithm, $\hat{\boldsymbol{\theta}}_{S_{1:M}} : \mathcal{Z}^k \to \Omega$. In the second stage, the learner computes $\hat{\boldsymbol{\theta}}_{S_{1:M}}(S_{M+1})$, the estimate of $\theta_{P_{M+1}}$.

The novel-task minimax risk is given by,

$$R_{\mathcal{P}}^*(\beta) = \inf_{f} \sup_{P_1, \ldots, P_{M+1} \in \mathcal{Q}_{\mathcal{P}}^{\beta}} \mathbb{E}_{\substack{S_{1:M} \sim P_{1:M}^n \\ S_{M+1} \sim P_{M+1}^k}} \left[ \ell\left( (f(S_{1:M}))(S_{M+1}), \theta_{P_{M+1}} \right) \right], \tag{2}$$

where $\mathcal{Q}_{\mathcal{P}}^{\beta} = \{(P_1, \ldots, P_{M+1}) \in \mathcal{P} : D_{\mathrm{KL}}(P_{M+1} \| P_i) \leq \beta, \text{ for } i = 1, \ldots, M\}$. This ensures a degree of relatedness between the novel and meta-training tasks.

The estimator for $\theta_{M+1}$ now depends additionally on the $Mn$ samples in $S_{1:M}$, where only $k \ll Mn$ samples from $P_{M+1}$ are available to the learner. Thus, $R_{\mathcal{P}}^*$ addresses the domain shift expected at

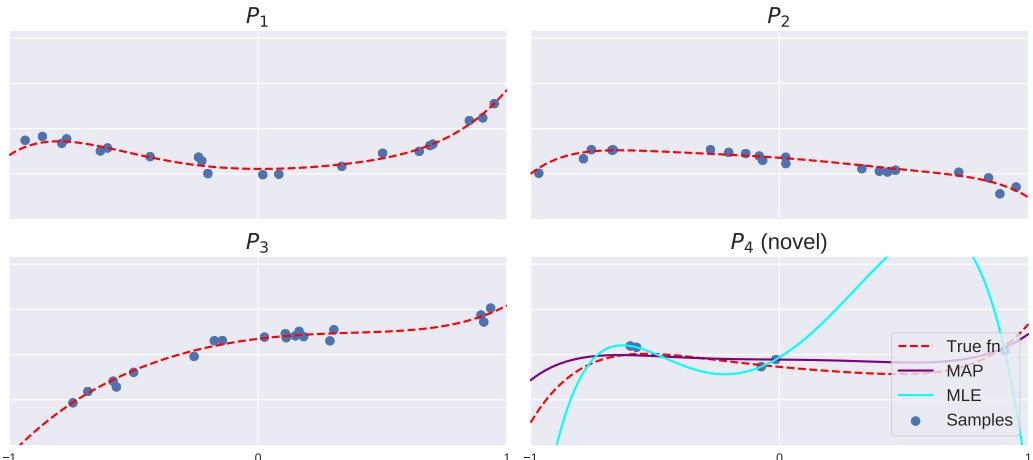

Figure 1: **Meta-learning 1D-regression:** The parameters of a 1D regression model are fitted from a small support set. The training distributions $(P_1, P_2, P_3)$ give a useful inductive bias for fitting $P_4$ using only 5 points. The MLE solution on the novel task for those 5 points is also displayed.

test-time in the meta-learning setting and allows the learner to use data from multiple tasks. The goal of $f$ is to learn an inductive bias from $S_{1:M}$ such that a good estimate is possible with only $k$ data points from $P_{M+1}$. In this setting, $k$ is equivalent to the number of shots in few-shot learning.

**An example with meta-linear regression**   We present here a short summary based on meta linear regression, which we will analyze in more detail in Section 5.

In Figure 1, we show observed data samples from a family of polynomial regression models. Our aim is to output an algorithm which recovers the parameters of a new polynomial function from limited observations–we choose a MAP estimator which is described fully in Section 5. In the bottom right, we are given only 5 data points from a novel task distribution and estimate the parameters of the model with both the MLE and MAP estimators — the MLE overfits the support set while the MAP estimator is close to the true function.

In terms of the terminology used above, the set,

$$\mathcal{P} = \{p_{\boldsymbol{\theta}}(y) = \mathcal{N}(\mathbf{x}^\top \boldsymbol{\theta}, \sigma^2) : \boldsymbol{\theta} \in \mathbb{R}^d, \mathbf{x} = [1, x, \ldots, x^{d-1}]\},$$

is the space of polynomial regression models, parameterized by $\boldsymbol{\theta}$. For this problem, we take $\ell(\hat{\theta}, \theta) = \|\hat{\theta} - \theta\|_2^2$. In Figure 1, tasks are generated with $p(\theta) = \mathcal{N}(\tau, \sigma_\theta^2)$, for unknown, sparse, $\tau \in \mathbb{R}^d$. Thus, each model is a polynomial function with few large coefficients. The algorithm $f$, first takes samples from $P_1, P_2, P_3$ and computes an estimate, $\hat{\tau}$. This estimate of $\tau$ is then used to compute $\hat{\boldsymbol{\theta}}(S_{M+1}; \hat{\tau}) = \mathrm{argmax}_{\boldsymbol{\theta}_4} \, p(\boldsymbol{\theta}_4 | \hat{\tau}, S_{M+1})$. Note that this approach is able to learn the correct inductive bias from the data, without requiring a carefully designed regularizer. The lower bounds we derive in Section 4 can be applied to problems of this general type, and the upper and lower bounds in Section 5 apply specifically to meta-learning linear regression.

## 4 INFORMATION THEORETIC LOWER BOUNDS ON NOVEL TASK GENERALIZATION

In this section, we first present our most general result: Theorem 1. Using this, we derive Corollary 1 that gives a lower bound in terms of the sample size in the training and novel tasks. Corollary 1 recovers a well-known *i.i.d* lower bound (Theorem 2) when $Mn = 0$, and, importantly, highlights that the novel task data is significantly more valuable than the training task data. Additionally, we provide a specialized bound that applies when the environment is partially observed — proving that in this setting training task data is insufficient to drive the minimax risk to zero.

In Theorem 1, we assume that $\mathcal{P}$ contains $J$ distinct $2\delta$-separated distributions but only $M + 1 \leq J$ tasks are visible to the learner. Intuitively, the error rate lower-bound shrinks as the amount of information shared between the training tasks and the novel task grows. All proofs are given in Appendix B.1. Recall $\ell(a, b) = \psi(\rho(a, b))$ for non-decreasing $\psi$ and arbitrary metric $\rho$.

**Theorem 1** (Minimax novel task risk lower bound). *Let $\mathcal{J} \subset \mathcal{P}$ contain $J$ distinct distributions such that $\rho(\theta_P, \theta_{P'}) \geq 2\delta$ and $D_{\mathrm{KL}}(P \| P') \leq \beta$ for all $P, P' \in \mathcal{J}$. Let $\pi$ be a random ordering of the $J$ elements, and $Z|\pi$ be a vector of $k$ i.i.d samples from $P_{\pi_{M+1}}$. Further, define $W|\pi$ to be an $n \times M$ matrix whose $j^{th}$ column consist of $n$ i.i.d samples from $P_{\pi_j}$. Then,*

$$R_{\mathcal{P}}^*(\beta) \geq \psi(\delta) \left( 1 - \frac{I(\pi_{M+1}; W) + I(\pi_{M+1}; Z) + 1}{\log_2 J} \right).$$

Note that $\delta$ is a property of the so-called packing set, $\mathcal{J}$, and may depend on the sample size, $\beta$, and other properties of $\mathcal{P}$. For example, practical instances of this bound typically require $\psi(\delta) = O(1/k)$ or similar, as in Theorem 3 below. To derive this result, we bound the statistical estimation error by the error on a corresponding decoding problem where we must predict the novel task index, given the meta-training set $S_{1:M}$ and $S_{M+1}$. Fano's inequality provides best-case error rates for this problem.

Using Theorem 1, we derive our first bound on the novel-task minimax risk that depends on the number of meta-training tasks ($M$) and datapoints per training task ($n$, $k$), via a local-packing argument. The following corollary implies that if we have $J$ meta-training tasks in our $2\delta$-packing that are close (in terms of their pairwise KL distance), then learning a novel task from training samples drawn from the meta-training tasks requires significantly more examples; in particular, learning the novel task from samples drawn from the meta training set requires $\Omega(J)$ times the sample complexity of the novel task. This matches our intuition that learning the novel task implies the ability to distinguish it from all $J$ well-separated meta-training tasks.

**Corollary 1.** *Assume the same setting as in Theorem 1. Then,*

$$R_{\mathcal{P}}^*(\beta) \geq \psi(\delta) \left( 1 - \frac{1 + \left( \frac{Mn}{(J-1)} + k \right) \frac{1}{J^2} \sum_{1 \leq i,j \leq J} D_{\mathrm{KL}}(P_i \| P_j)}{\log_2 J} \right).$$

**A tighter bound on partially observed environments** We now consider the special case of Theorem 1 when $M < J - 1$, meaning that the meta-training tasks cannot cover the full packing set. In this setting, we prove that no algorithm can generalize perfectly to tasks in unseen regions of the space with small $k$, regardless of the number of data points $n$ observed in each meta-training task.

**Corollary 2.** *Assume the same setting as in Theorem 1, with $M + 1 < J$. Then,*

$$R_{\mathcal{P}}^* \geq \psi(\delta) \left( \frac{\log_2(J - M) - \frac{k}{J^2} \sum_{1 \leq i,j \leq J} D_{\mathrm{KL}}(P_i \| P_j) - 1}{\log_2 J} \right).$$

In this work, we have focused on the setting where $W$ contains an equal number of samples from each of the meta-training tasks — this is the sampling scheme shown in Figure 2. However, it is possible to extend these results to different sampling schemes for $W$. For example, in the appendix we derive bounds with $W|\pi$ as a mixture distribution. Surprisingly, despite task identity being hidden from the learner, the asymptotic rate for these two sampling schemes match.

## 4.1 MEASURING TASK-RELATEDNESS

The use of local packing requires the design of an appropriate set of distributions whose corresponding parameters are $2\delta$-separated but maintain small KL divergences. In the multi-task setting such an assumption is intuitively reasonable: challenging tasks should require separated parameters for ideal explanations ($2\delta$-separated) but should satisfy some relatedness measure (small KL). Importantly, these parameters can depend on sample size and other problem-specific variables. As we will see shortly, lower bounds on minimax risk in the *i.i.d* setting may also assume the same notion of relatedness for the local-packing in $\mathcal{P}$.

Task relatedness is a necessary feature for upper-bounds on novel task generalization, but are typically difficult to define (see e.g. Ben-David & Borbely (2008)). Our lower bounds utilize a

relatively weak notion of task-relatedness, and thus may be overly pessimistic compared to the upper bounds computed in existing work. However, task relatedness of this form can be formulated in a representation space shared across tasks and thus can be applied in settings like those explored by e.g. Du et al. (2020). Deriving lower bounds under the different task relatedness assumptions present in the literature would make for exciting future work.

### 4.2 COMPARISON TO RISK OF *i.i.d* LEARNERS

From the statement of Theorem 1 it is not clear how this lower-bound compares to that of the *i.i.d* learner which has access only to the $k$ samples from $S_{M+1}$. To investigate the benefit of additional meta-training tasks, we compare our derived minimax risk lower bounds to those achieved by *i.i.d* learners. To do so, we revisit standard minimax lower bounds that can be found in e.g. Loh (2017).

**Theorem 2** (IID minimax lower-bound). *Suppose* $\{P_1, \ldots, P_J\} \subseteq \mathcal{P}$ *satisfy* $\rho(\theta_{P_i}, \theta_{P_j}) \geq 2\delta$ *for all* $i \neq j$. *Then,*

$$R^* \geq \psi(\delta) \left( 1 - \frac{\frac{k}{J^2} \sum_{1 \leq i,j \leq J} D_{\mathrm{KL}}\left(P_i \| P_j\right) + 1}{\log_2 J} \right).$$

We include a proof of this result in Appendix B.1, using local-packing as in our meta-learning bounds. As hoped, Corollary 1 recovers Theorem 2 when there are no training tasks available. Moreover, this *i.i.d* bound is strictly larger than the one computed in Corollary 1 in general. Note that while this *i.i.d* minimax risk is asymptotically tight for several learning problems (Loh, 2017; Raskutti et al., 2011), there is no immediate guarantee that the same is true for our meta-learning minimax bounds. We investigate the quality of these bounds by providing comparable upper bounds in the next section.

## 5 ANALYSIS OF A HIERARCHICAL BAYESIAN MODEL OF META-LEARNING

Our goal is to analyze the sample complexity of meta-learning for linear regression, where samples are drawn from multiple meta-training tasks and we want to generalize to a new task with only a few data points. After introducing the setting, we will compute lower-bounds on the minimax risk using our results from Section 4, revealing a $2^d$ scaling on the meta-training sample complexity. Following the lower bound, we derive an accompanying upper-bound on the risk of a MAP estimator, derived from an empirical Bayes estimate over a hierarchical Bayesian model. Asymptotic analysis of this bound reveals that if the observed samples from the novel task vary considerably more than the task parameters, then observing more meta-training samples may significantly improve convergence in the small $k$ regime. This is validated empirically in Section 6.

For $i = 1 \ldots M + 1$, where $M + 1$ is the total number of tasks, we define,

$$\mathbf{y}_i = X_i \boldsymbol{\theta}_i + \boldsymbol{\epsilon}_i, \qquad X_i \in \mathbb{R}^{n_i \times d}, \mathbf{y}_i \in \mathbb{R}^{n_i}, \boldsymbol{\epsilon}_i \in \mathbb{R}^{n_i}$$
$$\boldsymbol{\epsilon}_i \sim \mathcal{N}(0, \sigma_i^2 I), \qquad \sigma_i^2 \in \mathbb{R}^+$$

Each task has some design matrix $X_i$ and unknown parameters $\boldsymbol{\theta}_i$. For simplicity, we assume known isotropic noise models and that $n_i = n$ for all $i \leq M$, with $n_{M+1} = k$.

Our meta learner will fit the data using an empirical Bayes estimate in a hierarchical Bayesian model:

$$\boldsymbol{\theta}_i = \boldsymbol{\tau} + \boldsymbol{\xi}, \quad \boldsymbol{\tau} \in \mathbb{R}^d, \quad \boldsymbol{\xi} \in \mathbb{R}^d, \quad \boldsymbol{\xi} \sim \mathcal{N}(0, \sigma_\theta^2 I), \quad \sigma_\theta^2 \in \mathbb{R}^+$$

We will consider the Maximum a Posterior estimator,

$$\hat{\boldsymbol{\theta}}_{M+1} = \operatorname*{argmax}_{\boldsymbol{\theta}_{M+1}} p(\boldsymbol{\theta}_{M+1} | \mathbf{y}_1, \ldots, \mathbf{y}_{M+1}),$$

and will characterize its risk, $\mathbb{E}[\|\hat{\boldsymbol{\theta}}_{M+1} - \boldsymbol{\theta}_{M+1}\|_2^2]$, where the expectation is with respect to sampled data only. The posterior distribution under the Empirical Bayes estimate for $\boldsymbol{\tau}$ is given in Appendix C.2. The derivation is standard but dense and we recommend dedicated readers to consult Gelman et al. (2013), or an equivalent text, for more details.

## 5.1 MINIMAX LOWER BOUNDS

We now compute lower bounds for parameter estimation with meta-learning over multiple linear regression tasks. Beginning with a definition of the space of data generating distributions,

$$\mathcal{P}_{LR} = \{p_{\boldsymbol{\theta}}(\mathbf{y}) = \mathcal{N}(X\boldsymbol{\theta}, \sigma^2 I) : \boldsymbol{\theta} \in \mathbb{B}_2(1), X \in \mathbb{R}^{n \times d}.\}$$

where $\boldsymbol{\theta}$ are the parameters to be learned, and $X$ is the design matrix of each linear regression task in the environment. We write $\gamma = \max_i \sigma_{\max}(X_i/\sqrt{n})$, which we assume is bounded for all $X$ and $n$ (an assumption that is validated for random Gaussian matrices by Raskutti et al. (2011)).

**Theorem 3** (Meta linear regression lower bound). *Consider $\mathcal{P}_{LR}$ defined as above and let $\ell(a, b) = (\|a - b\|_2)^2$. If $d \geq 2$ and $2^{-d}M + kn^{-1} \geq \max\{\frac{d}{4\beta}, d\sigma^2/(256\gamma^2 n)$, then,*

$$R^*_{\mathcal{P}_{LR}}(\beta) \geq O\left(\frac{d\sigma^2}{\gamma^2(2^{-d}nM + k)}\right)$$

The proof is given in Appendix B.5. We see that the size of the meta-training set has an inverse exponential scaling in the dimension, $d$. This reflects the complexity of the space growing exponentially in dimensions and the need for a matching growth in data size to cover the environment sufficiently.

## 5.2 MINIMAX UPPER BOUNDS

To compute upper bounds on the estimation error, we require an additional assumption. Namely, we will assume that the design matrices also have bounded minimum singular values, $0 < s \leq \sigma_{\min}(X/\sqrt{n})$ (see Raskutti et al. (2011) for some justification). For the upper-bounds, we allow the bounds on the singular values of the design matrices and the observation noise in the novel task to be different than those in the meta-training tasks. We note that we can still recover the setting assumed in the lower bounds, where all tasks match on these parameters, as a special case.

The learner observes $n$ data points from each linear regression model in $\{P_{\boldsymbol{\theta}_1}, \ldots, P_{\boldsymbol{\theta}_M}\} \subset \mathcal{P}$. We then bound the error of estimating $\boldsymbol{\theta}_{M+1}$, for which $k$ samples are available.

The expected error rate of the MAP estimator can be decomposed as the posterior variance and bias squared. In the appendix we provide a detailed derivation of these results. The bound depends on dimensionality $d$, the observation noise in each task $\sigma_i^2$, the number of tasks $M$, the number of data points in each meta-training task $n$, and the number of data points in the novel task $k$.

**Theorem 4** (Meta Linear Regression Upper Bound). *Let $\hat{\boldsymbol{\theta}}_{M+1}$ be the maximum-a-posteriori estimator, $\mu_{\boldsymbol{\theta}_{M+1}|Y_{1:M+1}}$. Then,*

$$R^*_{\mathcal{P}_{LR}} \leq \sup_{\boldsymbol{\theta}_1, \ldots, \boldsymbol{\theta}_{M+1} \in \mathbb{B}_2(1)} \mathbb{E}[\|\hat{\boldsymbol{\theta}}_{M+1} - \boldsymbol{\theta}_{M+1}\|^2] \leq O\left(d\sigma^2_{M+1}C(M, n, k)^{-2}D(M, n, k)\right)$$

*where,*

$$C(M, n, k) = \left[k + \frac{Mn}{\frac{n(M+\kappa^2)s_2^2}{\alpha_2} + A}\right], \text{ and, } D(M, n, k) = \left[k + \frac{Mn}{(\frac{n}{L_1} + A_1)(\frac{Mn}{L_2} + A_2)}\right].$$

*Expectations are taken over the data conditioned on $\boldsymbol{\theta}_1, \ldots, \boldsymbol{\theta}_{M+1}$. Additional terms not depending on $d$, $M$, $n$, $k$ are defined in Appendix C.2.*

While the bounds presented in Theorem 4 are relatively complicated, we can probe the asymptotic convergence of the MAP estimator to the true task parameters, $\boldsymbol{\theta}_{M+1}$. In the following section, we will discuss some of the consequences of this result and its implications for our lower bounds.

## 5.3 ASYMPTOTIC BEHAVIOR OF THE MAP ESTIMATOR

We first notice that when $k$ is small, the risk cannot be reduced to zero by adding more meta-training data. Recent work has suggested such a relationship may be inevitable (Hanneke & Kpotufe, 2020). Our lower bound presented in Corollary 2 agrees that more samples from a small number of meta-training tasks will not reduce the error to zero. However, unlike our lower bounds based on local

packing, the lower bounds presented in this section predict that if the meta-training tasks cover the space sufficiently then an optimal algorithm might hope to reduce the error entirely with enough samples. We hypothesize that this gap is due to limitations in the standard proof techniques we utilize for the lower-bounds when the number of tasks grows, and expect a sharper bound may be possible.

To emulate the few-shot learning setting where $k$ is relatively small, we consider $n \to \infty$, with $k$ and $M$ fixed. In this case, the risk is bounded as,

$$\sup_{\boldsymbol{\theta}_1,\ldots,\boldsymbol{\theta}_{M+1} \in \mathbb{B}_2(1)} \mathbb{E}[\|\hat{\boldsymbol{\theta}}_{M+1} - \boldsymbol{\theta}_{M+1}\|^2] \leq O\left(d\sigma_{M+1}^2 \left[k + \frac{2\alpha_2 M}{M + \kappa^2}\right]^{-1}\right),$$

where $\alpha_2 = \sigma_{M+1}^2/\sigma_\theta^2$, is the ratio of the observation noise to the variance in sampling $\boldsymbol{\theta}$, and $\kappa$ is the condition number of the design matrices. This leads to a key takeaway: if the observed samples from $P_{M+1}$ vary considerably more than the parameters $\boldsymbol{\theta}$, then observing more samples in $S_{1:M}$ will significantly improve convergence towards the true parameters in the small $k$ regime. Further, adding more tasks (increasing $M$) also improves these constant factors by removing the dependence on the condition number, $\kappa$.

## 6 EMPIRICAL INVESTIGATIONS

In this section, we provide additional quantitative exploration of the upper bound studied in Section 5. The aim is to take steps towards relating the bounds to experimental results; we know of little theoretical work in meta-learning that attempt to relate their results to practical empirical datasets. Full details of the experiments in this section can be found in Appendix D.

### 6.1 HIERARCHICAL BAYES POLYNOMIAL REGRESSION

We first focus on the setting of polynomial regression over inputs in the range $[-1, 1]$. Some examples of these functions and samples are presented in Figure 1, alongside the MAP and MLE estimates for the novel task.

Figure 2 shows the analytical expected error rate (risk) under various environment settings. We observe that even in this simple hierarchical model, the estimator exhibits complex behavior that is correctly predicted by Theorem 4. In Figure 2A, we varied the novel task difficulty by increasing the novel task observation noise ($\sigma_{M+1}^2$). We plot three curves for three different dataset size configurations. When the novel task is much noisier than the source tasks, it is greatly beneficial to add more meta-training data (blue vs. red). And while larger $k$ made little difference when the novel task was relatively difficult (blue vs. green), the expected loss was orders of magnitude lower when the novel task became easier. In Figure 2B, we fixed the relative task difficulty and instead varied $k$ and $M$. The x-axis now indicates the total data $Mn + k$ available to the learner. We observed that adding more tasks has a large effect in the low-data regime but, as predicted, the error has a non-zero asymptotic lower-bound — eventually it is more beneficial to add more novel-task data samples.

These empirical simulations verify that our theoretical analysis is predictive of the behavior of this meta learning algorithm, as each of these observations can be inferred from Theorem 4. While this model is simple, it captures key features of and provides insight into the more general meta-learning problem. We also explored a non-linear sinusoid meta-regression problem (Finn et al., 2017), finding that our theory is largely predictive of the general trends in this setting too.

### 6.2 SINUSOID REGRESSION WITH MAML

Following the connections between MAML and hierarchical Bayes explored by Grant et al. (2018), we also explored regression on sinusoids using MAML. Our aim was to investigate how predictive our linear theory is for this highly non-linear problem setting. As in Finn et al. (2017), we sample sinusoid functions by placing a prior over the amplitude and phase. In other works (Finn et al., 2017; Grant et al., 2018) the same prior is used for the training and testing stages. However, to better measure generalization to novel tasks we use different prior distributions when training versus evaluating the model.

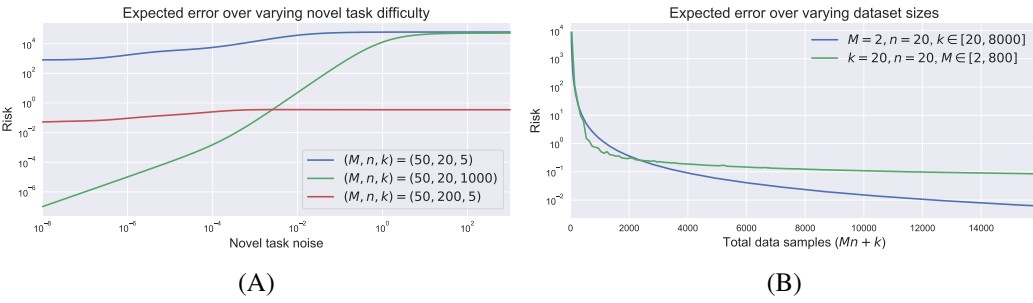

(A)  (B)

Figure 2: The expected error rate of the hierarchical MAP estimator, $\hat{\boldsymbol{\theta}}_{M+1}$, over different environment hyperparameter settings. **A)** The novel task observation noise is increased, making the novel task harder to learn. **B)** We increase the size of the dataset, in one case adding new tasks ($M$) and in the other adding new novel task data samples ($k$).

We display the risk averaged over 30 trials in Figure 3. We varied the novel task difficulty by increasing the observation noise in the novel task. We plot separate curves for different dataset size configurations, and observe that the empirical results align fairly well with the results derived by sampling the hierarchical model (Figure 2A). Adding more meta-training data (increasing $n$) is beneficial (green vs. yellow) and adding more test data-points (higher $k$) is also beneficial (red vs. green). Here however, these relationships did not interact with the task difficulty, as the wins for increased meta-training and meta-testing data were consistent, until task noise prevents any setting of the model from performing the task.

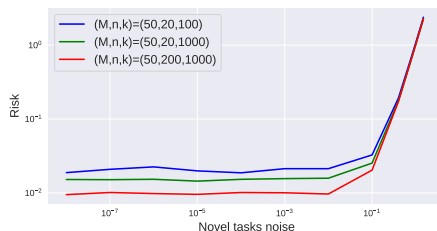

Figure 3: Average risk for regressing sinusoid functions with MAML.

## 7  CONCLUSION

Meta-learning algorithms identify the inductive bias from source tasks and make models more adaptive towards unseen novel distribution. In this paper, we take initial steps towards characterizing the difficulty of meta-learning and understanding how these limitations present in practice. We have derived both lower bounds and upper bounds on the error of meta-learners, which are particularly relevant in the few-shot learning setting where $k$ is small. Our bounds capture key features of the meta-learning problem, such as the effect of increasing the number of shots or training tasks. We have also identified a gap between our lower and upper bounds when there are a large number of training tasks, which we hypothesize is a limitation of the proof technique that we applied to derive the lower bounds — suggesting an exciting direction for future research.

## 8  ACKNOWLEDGEMENTS

This work benefited greatly from the input of many other researchers. In particular, we extend our thanks to Shai Ben-David, Karolina Dziugaite, Samory Kpotufe, and Daniel Roy for discussions and feedback on the results presented in this work. We thank Ahmad Beirami and anonymous reviewers for their valuable feedback that led to significant improvements to this paper. We also thank Elliot Creager, Will Grathwohl, Mufan Li, and many of our other colleagues at the Vector Institute for feedback that greatly improved the presentation of this work. Resources used in preparing this research were provided, in part, by the Province of Ontario, the Government of Canada through CIFAR, and companies sponsoring the Vector Institute (`www.vectorinstitute.ai/partners`).

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

# A    NOTATION

| | Description |
|---|---|
| $\mathcal{X}$ | The domain of the data, e.g. $\mathbb{R}^d$ |
| $\mathcal{Y}$ | The range of the data, e.g. $\mathbb{R}$ |
| $\mathcal{Z}$ | The product space $\mathcal{X} \times \mathcal{Y}$ |
| $\mathcal{P}$ | A collection of distributions over $\mathcal{Z}$ |
| $\mathcal{J}$ | A (finite) subset of distributions in $\mathcal{P}$ |
| $P$ | An element of $\mathcal{P}$ |
| $P^k$ | The product distribution, whose samples correspond to $k$ independent draws from $P$ |
| $P_{1:M}$ | The product distribution, $\Pi_{i=1}^{M} P_i$, for $P_i \in \mathcal{P}$ |
| $\bar{P}_{1:M}$ | The mixture distribution, $\frac{1}{M} \sum_{i=1}^{M} P_i$, for $P_i \in \mathcal{P}$ |
| $\Omega$ | A metric space, containing parameters for each distribution |
| $\theta$ | A functional, mapping distributions in $\mathcal{P}$ to parameters in $\Omega$ |
| $\hat{\theta}$ | An estimator $\hat{\theta} : \mathcal{Z}^n \to \mathcal{F}$ |
| $I(X;Y)$ | The mutual information between random variables $X$ and $Y$. |
| $S, S_{M+1}$ | Denotes training datasets drawn *i.i.d* from some $P \in \mathcal{P}$. Typically $S = \{z_1, \ldots, z_n\}$, $S_{M+1} = \{z'_1, \ldots, z'_k\}$ |
| $S_{\mathcal{P}}$ | Denotes a meta-training set drawn *i.i.d* from $P_{1:M}$ |
| $[N]$, for $N \in \mathbb{N}$ | Indicates the set $\{1, \ldots, N\}$ |
| $\mathbb{B}_p(r)$ | The $p$-norm ball of radius $r$, centered at 0. |

Table 1: Summary of notation used in this manuscript

# B    LOWER BOUND PROOFS

We will make use of several standard results below, which we present here.

**Lemma 1.** *Fano's Inequality (Fano, 1961; Cover & Thomas, 2012)  For any estimator $\hat{Y}$ of a random variable $Y$ such that $Y \to Z \to \hat{Y}$ forms a Markov chain, it holds that,*

$$\mathbb{P}(\hat{Y} \neq Y) \geq \frac{H(Y|Z) - 1}{\log_2 |Y|} = \frac{H(Y) - I(Y;Z) - 1}{\log_2 |Y|}.$$

**Lemma 2.** *Mutual information equality (Khas' minskii, 1979) Consider random variables $Z_1, Z_2, Y$, then,*

$$I(Y;(Z_1, Z_2)) + I(Z_1; Z_2) = I(Z_1;(Z_2, Y)) + I(Z_2;Y)$$

**Lemma 3.** *Local packing lemma (Loh, 2017) Consider distributions $P_1, \ldots, P_J \in \mathcal{P}$. Let $Y$ be a random variable distributed uniformly on $[J]$ and let $Z|\{Y = j\}$ be a vector of $k$ i.i.d samples from*

$P_j$. Then,

$$I(Y;Z) \leq \frac{k}{J^2} \sum_{1 \leq i,j \leq J} D_{\mathrm{KL}}(P_i \| P_j).$$

We will require a novel local packing bound for the novel-task risk, which we present in Lemma 5.

## B.1 IID LOWER BOUND

We first prove the *i.i.d* result, which will serve as a guide for our novel lower bounds.

**Theorem 2** (IID minimax lower-bound). *Suppose* $\{P_1, \ldots, P_J\} \subseteq \mathcal{P}$ *satisfy* $\rho(\theta_{P_i}, \theta_{P_j}) \geq 2\delta$ *for all* $i \neq j$. *Then,*

$$R^* \geq \psi(\delta) \left( 1 - \frac{\frac{k}{J^2} \sum_{1 \leq i,j \leq J} D_{\mathrm{KL}}(P_i \| P_j) + 1}{\log_2 J} \right).$$

*Proof.* First, notice that,

$$\sup_{P \in \mathcal{P}} \mathbb{E}_{S \sim P^k} \left[ \ell(\hat{\theta}(S), \theta_P) \right] \geq \frac{1}{J} \sum_{i=1}^{J} \mathbb{E}_{S \sim P_i^k} \left[ \ell(\hat{\theta}(S), \theta_{P_i}) \right].$$

Now define the decision rule,

$$f(S) = \operatorname*{argmin}_{1 \leq j \leq J} \rho(\hat{\theta}(S), \theta_{P_j})],$$

with ties broken arbitrarily. We proceed by bounding the expected loss. First, using Markov's inequality,

$$\mathbb{E}_{S \sim P_i^k} \left[ \ell(\hat{\theta}(S), \theta_{P_i}) \right] \geq \psi(\delta) \mathbb{P}_{S \sim P_i^k} \left[ \psi(\rho(\hat{\theta}(S), \theta_{P_i})) \geq \psi(\delta) \right],$$
$$= \psi(\delta) \mathbb{P}_{S \sim P_i^k} \left[ \rho(\hat{\theta}(S), \theta_{P_i}) \geq \delta \right].$$

Next, consider the case $\rho(\hat{\theta}(S), \theta_i)) < \delta$. Through the triangle inequality,

$$\rho(\hat{\theta}(S), \theta_{P_j}) \geq \rho(\theta_{P_i}, \theta_{P_j}) - \rho(\hat{\theta}(S), \theta_{P_i})$$
$$\geq 2\delta - \delta > \rho(\hat{\theta}(S), \theta_{P_i})$$

Thus, the probability that the distance is less than $\delta$ is at least as large as the probability that the estimator is correct.

$$\psi(\delta) \mathbb{P}_{S \sim P_i^k} \left[ \ell(\hat{\theta}(S), \theta_{P_i}) \geq \psi(\delta) \right] \geq \psi(\delta) \mathbb{P}(f(S) \neq i).$$

Now, using Fano's inequality with $Y = \pi_{M+1}$, and $\hat{Y} = f(S)$ (and the corresponding Markov chain $\pi_{M+1} \to S \to f(S)$), we have,

$$\frac{1}{J} \sum_{i=1}^{J} \mathbb{P}(f(S) \neq i) \geq \frac{\log_2 J - I(\pi_{M+1}; Z) - 1}{\log_2 J}.$$

Combining the above inequalities with the Local Packing Lemma gives the final result. $\square$

## B.2 PROOF OF THEOREM 1

**Theorem 1** (Minimax novel task risk lower bound). *Let* $\mathcal{J} \subset \mathcal{P}$ *contain* $J$ *distinct distributions such that* $\rho(\theta_P, \theta_{P'}) \geq 2\delta$ *and* $D_{\mathrm{KL}}(P \| P') \leq \beta$ *for all* $P, P' \in \mathcal{J}$. *Let* $\pi$ *be a random ordering of the* $J$ *elements, and* $Z | \pi$ *be a vector of* $k$ i.i.d *samples from* $P_{\pi_{M+1}}$. *Further, define* $W | \pi$ *to be an* $n \times M$ *matrix whose* $j^{th}$ *column consist of* $n$ i.i.d *samples from* $P_{\pi_j}$. *Then,*

$$R_{\mathcal{P}}^*(\beta) \geq \psi(\delta) \left( 1 - \frac{I(\pi_{M+1}; W) + I(\pi_{M+1}; Z) + 1}{\log_2 J} \right).$$

*Proof.* As in the *i.i.d* case, we first bound the supremum from below with an average,

$$\sup_{P'_1,\dots,P'_{M+1}\in\mathcal{P}} \mathbb{E}_{\substack{S_{1:M}\sim(P'_{1:M})^n \\ S_{M+1}\sim(P'_i)^k}}\left[\ell(\hat{\theta}_{S_{1:M}}(S_{M+1}),\theta_{(P'_i)})\right] \geq \frac{1}{J}\sum_{i=1}^{J}\frac{1}{\binom{J-1}{M}}\sum_{\pi|\{\pi_{M+1}=i\}}\mathbb{E}_{\substack{w\sim W|\pi \\ z\sim Z|\pi}}\left[\ell(\hat{\theta}_w(z),\theta_i)\right],$$

where the inner sum is over all length $M$ orderings, $\pi$ with $\pi_{M+1}=i$.

As before, we consider the following estimator,

$$f(W,Z) = \underset{1\leq j\leq J}{\operatorname{argmin}}\,\rho(\hat{\theta}_W(Z),\theta(P_j))$$

Using Markov's inequality, and then following the proof of Theorem 2, we have,

$$\frac{1}{J}\sum_{i=1}^{J}\frac{1}{\binom{J-1}{M}}\sum_{\pi|\{\pi_{M+1}=i\}}\mathbb{E}_{\substack{w\sim W|\pi \\ z\sim Z|\pi}}\left[\ell(\hat{\theta}_w(z),\theta_i)\right] \geq \frac{1}{J}\sum_{i=1}^{J}\frac{1}{\binom{J-1}{M}}\sum_{\pi|\{\pi_{M+1}=i\}}\psi(\delta)\mathbb{P}[f(W,Z)\neq i|\pi]$$

$$= \psi(\delta)\mathbb{P}[f(W,Z)\neq\pi_{M+1}]$$

with the use of Fano's inequality, we arrive at,

$$\psi(\delta)\left(1-\frac{I(\pi_{M+1};(W,Z))+1}{\log_2 J}\right)$$

Conditioned on $Y$, each element of $W$ and $Z$ are independent but they are not identically distributed. Thus, with the application of Lemma 2,

$$I(\pi_{M+1};(W,Z)) \leq I(\pi_{M+1};Z) + I(\pi_{M+1};W)$$

The result follows by combining these inequalities. $\qquad\square$

**Remark** In the above proof of Theorem 1, we did not need to make use of the form of the distribution of $W|Y=i$, only that the correct graph structure was observed. This grants us some flexibility, which we utilize later in Section B.4 to prove lower bounds for mixture distributions.

We now proceed with proofs of the corollaries of Section 4.

### B.3   LOCAL PACKING RESULTS

**Lemma 4** (Meta-learning local packing). *Consider the same setting as in Theorem 1, then*

$$I(\pi_{M+1};W) \leq \frac{Mn}{J^2(J-1)}\sum_{1\leq i,j\leq J}D_{\mathrm{KL}}\left(P_i\|P_j\right)$$

*Proof.* There are $(J-1)!/(J-M-1)!$ orderings on the first $M$ indices, given the $(M+1)^{th}$. We introduce the following notation,

$$\bar{P}_{-i} := \frac{(J-M-1)!}{(J-1)!}\sum_{\pi|\pi_{M+1}=i}p(W|\pi) \qquad\qquad \bar{P} := \frac{1}{J}\sum_{i=1}^{J}\bar{P}_{-i}$$

As in previous proofs, we notice that we can write,

$$\bar{P} = \frac{1}{J}\bar{P}_{-i} + \frac{J-1}{J}\frac{1}{J-1}\sum_{j\neq i}\bar{P}_{-j}$$

First, note that we can upper bound $I(\pi_{M+1}; W) \leq nI(\pi_{M+1}; w)$, where $w$ denotes a single row in $W$. Further,

$$
\begin{aligned}
I(\pi_{M+1}; w) &= \frac{1}{J} \sum_{i=1}^{J} \mathbb{E} \log \frac{\bar{P}_{-i}}{\bar{P}} \\
&= \frac{1}{J} \sum_{i=1}^{J} D_{\mathrm{KL}} \left( \bar{P}_{-i} \| \frac{1}{J} \bar{P}_{-i} + \frac{J-1}{J} \frac{1}{J-1} \sum_{j \neq i} \bar{P}_{-j} \right) \\
&\leq \frac{1}{J} \sum_{i=1}^{J} \frac{1}{J} D_{\mathrm{KL}} \left( \bar{P}_{-i} \| \bar{P}_{-i} \right) + \frac{J-1}{J} D_{\mathrm{KL}} \left( \bar{P}_{-i} \| \frac{1}{J-1} \sum_{j \neq i} \bar{P}_{-j} \right) \\
&\leq \frac{1}{J(J-1)} \sum_{1 \leq i \neq j \leq J} D_{\mathrm{KL}} \left( \bar{P}_{-i} \| \bar{P}_{-j} \right)
\end{aligned}
$$

We will use convexity of the KL divergence to upper bound this quantity. Each distribution $\bar{P}_{-i}$ is an average over a random selection of index orderings.

When applying convexity, all pairs of selections that exactly match will lead to a KL divergence of zero. There are the same number of these in each component of $\bar{P}_{-i}$. Thus we care only about selections that contain either $j$ or $i$ such that matching pairs of distributions exactly is not possible. Further, we need only consider pairs of product distributions who differ only in a single, identical position.

Each of the above described pairs of distributions has KL divergence equal to $D_{\mathrm{KL}} \left( P_j \| P_i \right)$. We conclude by counting the total number of orderings producing such pairs. First, there are $M$ choices for the index of $P_j$ and $P_i$. Then, there are $(J-2)!/(J-M-1)!$ total orderings of the remaining $M-1$ elements. Thus, we have,

$$
\begin{aligned}
I(\pi_{M+1}; w) &\leq \frac{1}{J(J-1)} \sum_{1 \leq i \neq j \leq J} D_{\mathrm{KL}} \left( \bar{P}_{-i} \| \bar{P}_{-j} \right) \\
&\leq \frac{1}{J(J-1)} \frac{(J-M-1)!}{(J-1)!} \frac{M(J-2)!}{(J-M-1)!} \sum_{1 \leq i \neq j \leq J} D_{\mathrm{KL}} \left( P_j \| P_i \right) \\
&= \frac{M}{J(J-1)^2} \sum_{1 \leq i \neq j \leq J} D_{\mathrm{KL}} \left( P_j \| P_i \right)
\end{aligned}
$$

$\square$

These results together provide an immediate proof of Corollary 1.

**Corollary 1.** *Assume the same setting as in Theorem 1. Then,*

$$
R_{\mathcal{P}}^*(\beta) \geq \psi(\delta) \left( 1 - \frac{1 + \left( \frac{Mn}{(J-1)} + k \right) \frac{1}{J^2} \sum_{1 \leq i,j \leq J} D_{\mathrm{KL}} \left( P_i \| P_j \right)}{\log_2 J} \right).
$$

*Proof.* Putting together the results of Theorem 1, Lemma 3, and Lemma 4, and using the fact that $D_{\mathrm{KL}} \left( P_i \| P_j \right) \leq \alpha$, the result follows immediately. $\square$

**Corollary 2.** *Assume the same setting as in Theorem 1, with $M + 1 < J$. Then,*

$$
R_{\mathcal{P}}^* \geq \psi(\delta) \left( \frac{\log_2(J-M) - \frac{k}{J^2} \sum_{1 \leq i,j \leq J} D_{\mathrm{KL}} \left( P_i \| P_j \right) - 1}{\log_2 J} \right).
$$

*Proof.* This result follows as an application of the data processing inequality. Notice that $\pi_{M+1} \to \pi_{1:M} \to W$ forms a Markov chain. Thus,

$$I(\pi_{M+1}; W) \leq I(\pi_{M+1}; \pi_{1:M}),$$

by the data processing inequality. We can compute $I(\pi_{M+1}; \pi_{1:M})$ in closed form:

$$I(\pi_{M+1}; \pi_{1:M}) = \log \frac{J}{J-M}.$$

The proof is completed by plugging in the *i.i.d* local packing bound alongside the above. □

### B.4 BOUNDS USING MIXTURE DISTRIBUTIONS

In this section we introduce tools to lower bound the minimax risk when the meta-training set is sampled from a mixture over the meta-training tasks, $\bar{P}_{1:M} = \frac{1}{M} \sum_{i=1}^{M} P_i$. We note first that Theorem 1 can be reproduced exactly when $W \sim \bar{P}_{1:M}$. Thus, we need only provide a local packing bound for the mixture distribution. In Lemma 5 we provide such a lower bound for the special case where $M = J - 1$, so that data is sampled from a mixture over the entire environment.

**Lemma 5** (Leave-one-task-out mixture local packing). *Let $\mathcal{J} \subset \mathcal{P}$ contain $J$ distinct distributions such that $\rho(\theta_P, \theta_{P'}) \geq 2\delta$ for all $P, P' \in \mathcal{J}$ and let $\bar{P}_{-i} = \frac{1}{J-1} \sum_{j \neq i} P_j$. Let $\pi$ be a random ordering of the $J$ elements, and define $W|\pi$ to be a vector of $n$ i.i.d samples from $\bar{P}_{-\pi_{M+1}}$. Then,*

$$I(\pi_{M+1}; W) \leq \frac{1}{(J-1)J^2} \sum_{1 \leq i,j \leq J} D_{\mathrm{KL}}\left(P_i \| P_j\right).$$

*Proof.* From Lemma 3 (and some simple arithmetic) we have,

$$I(\pi_{M+1}; W) = \frac{1}{J} \sum_{i=1}^{J} D_{\mathrm{KL}}\left(\bar{P}_{-i} \| \bar{P}_{1:J}\right).$$

Note that by the definition of the mixture distribution,

$$\bar{P}_{1:J} = \frac{J-1}{J} \bar{P}_{-i} + \frac{1}{J} P_i.$$

Using the convexity of the KL divergence,

$$\begin{aligned}
I(\pi_{M+1}; W) &= \frac{1}{J} \sum_{i=1}^{J} D_{\mathrm{KL}}\left(\bar{P}_{-i} \Big\| \frac{J-1}{J} \bar{P}_{-i} + \frac{1}{J} P_i\right) \\
&\leq \frac{1}{J} \sum_{i=1}^{J} \frac{J-1}{J} D_{\mathrm{KL}}\left(\bar{P}_{-i} \| \bar{P}_{-i}\right) + \frac{1}{J} D_{\mathrm{KL}}\left(\bar{P}_{-i} \| P_i\right) \\
&= \frac{1}{J^2} \sum_{i=1}^{J} D_{\mathrm{KL}}\left(\bar{P}_{-i} \| P_i\right) \\
&= \frac{1}{J^2} \sum_{i=1}^{J} D_{\mathrm{KL}}\left(\frac{1}{J-1} \sum_{1 \leq j \leq J, j \neq i} P_j \Big\| P_i\right) \\
&\leq \frac{1}{(J-1)J^2} \sum_{i=1}^{J} \sum_{1 \leq j \leq J, j \neq i} D_{\mathrm{KL}}\left(P_j \| P_i\right) \\
&= \frac{1}{(J-1)J^2} \sum_{1 \leq i,j \leq J} D_{\mathrm{KL}}\left(P_i \| P_j\right).
\end{aligned}$$

Noting for the last step that the KL is zero if and only if the distributions are the same almost everywhere. □

### B.5 Proof of Hierarchical linear model lower bound

Recall that the space of distributions we consider is given by,

$$\mathcal{P}_{LR} = \{p_{\boldsymbol{\theta}}(\mathbf{y}) = \mathcal{N}(X\boldsymbol{\theta}, \sigma^2 I) : \boldsymbol{\theta} \in \mathbb{B}_2(1), X \in \mathbb{R}^{n \times d}.\}$$

**Theorem 3** (Meta linear regression lower bound). *Consider $\mathcal{P}_{LR}$ defined as above and let $\ell(a,b) = (\|a-b\|_2)^2$. If $d \geq 2$ and $2^{-d}M + kn^{-1} \geq \max\{\frac{d}{4\beta}, d\sigma^2/(256\gamma^2 n)$, then,*

$$R^*_{\mathcal{P}_{LR}}(\beta) \geq O\left(\frac{d\sigma^2}{\gamma^2(2^{-d}nM + k)}\right)$$

*Proof.* The proof consists of two steps, we first construct a $2\delta$-packing of $\mathcal{P}_{LR}$. Then, we upper bound the KL divergence between two distributions in this packing and use Corollary 1 to give the desired bound.

The maximal packing number $J$ for the unit 2-norm ball can be bounded by the following,

$$\left(\frac{1}{\delta}\right)^d \leq J \leq \left(1 + \frac{2}{\delta}\right)^d.$$

We use a common scaling trick. First, through this bound, we can build a packing set, $\mathcal{V}$, with packing radius $1/2$, giving $2^d \leq J \leq 5^d$. We define a new packing set of the same cardinality by taking $\theta_i = 4\delta v_i$ for all $v_i \in \mathcal{V}$ (requiring $\delta \leq 1/2$). Giving for all $i \neq j$,

$$\|\theta_i - \theta_j\| = 4\delta\|v_i - v_j\| \geq 2\delta$$

similarly, $\|\theta_i - \theta_j\| \leq 4\delta$.

We now proceed with bounding the KL divergences.

$$\begin{aligned}
D_{\mathrm{KL}}\left(P_i\|P_j\right) &= \frac{1}{2\sigma^2}\|X_i\boldsymbol{\theta}_i - X_j\boldsymbol{\theta}_j\|_2^2 \\
&= \frac{1}{2\sigma^2}\left(\boldsymbol{\theta}_i^\top X_i^\top X_i\boldsymbol{\theta}_i + \boldsymbol{\theta}_j^\top X_j^\top X_j\boldsymbol{\theta}_j - 2\boldsymbol{\theta}_i^\top X_i^\top X_j\boldsymbol{\theta}_j\right) \\
&\leq \frac{1}{2\sigma^2}\left(n_i\gamma_i^2\|\boldsymbol{\theta}_i\|^2 + n_j\gamma_j^2\|\boldsymbol{\theta}_j\|^2 - 2\boldsymbol{\theta}_i^\top X_i^\top X_j\boldsymbol{\theta}_j\right)
\end{aligned}$$

where $\gamma_i^2 = \sup_{\boldsymbol{\theta}} \frac{\|X_i\boldsymbol{\theta}\|}{\sqrt{n_i}\|\boldsymbol{\theta}\|}$. We write $n = \max_k n_k$ and $\gamma = \max_k \gamma_k$, then,

$$\begin{aligned}
D_{\mathrm{KL}}\left(P_i\|P_j\right) &\leq \frac{n\gamma^2}{2\sigma^2}\left(\|\boldsymbol{\theta}_i\|^2 + \|\boldsymbol{\theta}_j\|^2 - \frac{2}{n\gamma^2}\boldsymbol{\theta}_i^\top X_i^\top X_j\boldsymbol{\theta}_j\right) \\
&\leq \frac{n\gamma^2}{2\sigma^2}\left(\|\boldsymbol{\theta}_i\|^2 + \|\boldsymbol{\theta}_j\|^2 + 2\|\boldsymbol{\theta}_i\|\|\boldsymbol{\theta}_j\|\right) \\
&= \frac{n\gamma^2}{2\sigma^2}\left(\|\boldsymbol{\theta}_i\| + \|\boldsymbol{\theta}_j\|\right)^2 \leq \frac{32n\gamma^2\delta^2}{\sigma^2} \leq \beta
\end{aligned}$$

The second line is derived using the Cauchy-Schwarz inequality, and the final inequality uses $\|\boldsymbol{\theta}_i\| = \|4\delta v_i\| \leq 4\delta$. We will not proceed by invoking Corollary 1 on this packing set. This will require choosing $\delta$ to achieve our desired rate and will in turn impose constraints on the problem dimensions to ensure the packing is valid.

Now, using Corollary 1,

$$R^*_{\mathcal{P}_{LR}} \geq \delta^2\left(1 - \frac{(nM2^{-d} + k)32\gamma^2\delta^2/\sigma^2 + 1}{d}\right),$$

Choosing $\delta^2 = d\sigma^2 / \left[128\gamma^2(2^{-d}Mn + k)\right]$ gives,

$$1 - \frac{(nM2^{-d} + k)32\delta^2/\sigma^2 + 1}{d} = 1 - \frac{d/4 + 1}{d} \geq 1/4,$$

for $d \geq 2$. To enforce $\delta \leq 1/2$, we further require that,

$$2^{-d}Mn + k \geq \frac{d\sigma^2}{256\gamma^2}$$

Additionally, we may only consider packing sets with KL divergence no more than $\beta$, hence we also require that,

$$2^{-d}M + kn^{-1} \geq \frac{d}{4\beta}$$

Thus,

$$R_{\mathcal{P}}^* \geq O\left(\frac{d\sigma^2}{\gamma^2(2^{-d}nM + k)}\right)$$

$\square$

## C HIERARCHICAL BAYESIAN LINEAR REGRESSION UPPER BOUNDS

### C.1 SOME USEFUL LINEAR ALGEBRA RESULTS

Let $s_{\max}(A)$ denotes the maximum singular value of $A$; $s_{\min}(A)$ denotes the minimum singular value of $A$.

**Lemma 6.** *Singular value of sum of two matrices Let* $A, B \in \mathbb{R}^{m \times n}$, *then* $s_{\max}(A) + s_{\max}(B) \geq s_{\max}(A + B)$. *Furthermore, if* $A, B$ *are positive definite,* $s_{\min}(A) + s_{\min}(B) \leq s_{\min}(A + B)$.

*Proof.* The first result follows immediately from the triangle inequality of the matrix norm $\|\cdot\|_2$.

For the second result, suppose that $A$ and $B$ are positive definite.

$$s_{\min}(A + B) = \inf_{\|u\|=1} \|(A + B)u\| \tag{3}$$

$$= \sqrt{\inf_{\|u\|=1} \|(A + B)u\|^2} \tag{4}$$

$$= \sqrt{\inf_{\|u\|=1} \|Au\|^2 + \|Bu\|^2 + 2\langle Au, Bu \rangle} \tag{5}$$

$$= \sqrt{\inf_{\|u\|=1} \|Au\|^2 + \|Bu\|^2 + 2u^\top A^\top Bu} \tag{6}$$

$$\tag{7}$$

Now, notice that $A^\top B$ is similar to the matrix $A^{1/2}BA^{1/2}$, which exists as $A$ is positive definite. This matrix is itself positive definite, and thus has non-negative eigenvalues, meaning $A^\top B$ also has all positive eigenvalues. Thus, $u^\top A^\top Bu \geq 0$, for all $u$, and,

$$s_{\min}(A + B) \geq \sqrt{\inf_{\|u\|=1} \|Au\|^2 + \|Bu\|^2} \tag{8}$$

$$\geq \sqrt{\inf_{\|u\|=1} \|Au\|^2 + \inf_{\|v\|=1} \|Bv\|^2} \tag{9}$$

$$\geq \sqrt{s_{\min}^2(A) + s_{\min}^2(B)} \tag{10}$$

$$\geq s_{\min}(A) + s_{\min}(B) \qquad \text{(Concavity of } \sqrt{\cdot}) \tag{11}$$

$\square$

**Lemma 7.** *Singular value of product of two matrices Let* $A, B \in \mathbb{C}^{n \times n}$, *then* $s_{\max}(A)s_{\max}(B) \geq s_{\max}(AB)$, *and,* $s_{\min}(A)s_{\min}(B) \leq s_{\min}(AB)$.

First we prove the maximum singular value.

*Proof.*

$$s_{\max}(AB) = \sup_{\|v\|=1} \sqrt{v^* B^* A^* AB v} \tag{12}$$

$$= \sup_{\|v\|=1} \sqrt{\|Bv\|^2 u^* A^* Au} \text{ for } u = \frac{Bv}{\|Bv\|}, \tag{13}$$

$$\leq \sup_{\|v\|=1, \|u\|=1} \sqrt{\|Bv\|^2 u^* A^* Au} \tag{14}$$

$$= \sqrt{\sup_{\|v\|=1} \|Bv\|^2 \sup_{\|u\|=1} \|Au\|^2} \tag{15}$$

$$= \sqrt{s_{\max}{}^2(B) s_{\max}{}^2(A)} \tag{16}$$

$$= s_{\max}(A) s_{\max}(B). \tag{17}$$

The minimum singular value follows a similar structure. Suppose $AB$ is full rank,

$$s_{\min}(AB) = \inf_{\|v\|=1} \sqrt{v^* B^* A^* AB v} \tag{18}$$

$$= \inf_{\|v\|=1} \sqrt{\|Bv\|^2 u^* A^* Au} \text{ for } u = \frac{Bv}{\|Bv\|}, \tag{19}$$

$$\geq \inf_{\|v\|=1, \|u\|=1} \sqrt{\|Bv\|^2 u^* A^* Au} \tag{20}$$

$$= \sqrt{\inf_{\|v\|=1} \|Bv\|^2 \inf_{\|u\|=1} \|Au\|^2} \tag{21}$$

$$= \sqrt{s_{\min}{}^2(B) s_{\min}{}^2(A)} \tag{22}$$

$$= s_{\min}(A) s_{\min}(B). \tag{23}$$

If $AB$ is not full rank, then $s_{\min}(AB) = s_{\min}(A) s_{\min}(B) = 0$. □

**Lemma 8.** *(**Von Neumann's Trace Inequality** (Von Neumann, 1937)) Given two $n \times n$ complex matrices $A, B$, with singular vales $a_1 \geq \ldots \geq a_n$ and $b_1 \geq \ldots \geq b_n$ respectively. We have,*

$$|\mathrm{Tr}(AB)| \leq \sum_{i=1}^{n} a_i b_i$$

This is a classic result whose proof we exclude.

As a direct consequence of Lemma 8, $|\mathrm{Tr}(AB)| \leq n a_1 b_1$.

### C.2 POSTERIOR ESTIMATE

For reference, we reproduce the posterior estimate for the true parameters $\boldsymbol{\theta}_{M+1}$. As a shorthand, we write $Y_{1:M+1} = (\mathbf{y}_1, \ldots, \mathbf{y}_{M+1})$.

$$p(\boldsymbol{\tau}|Y_{1:M}) = \mathcal{N}(\mu_{\boldsymbol{\tau}|Y_{1:M}}, \Sigma_{\boldsymbol{\tau}|Y_{1:M}}), \tag{24}$$

$$\Sigma_{\tau|Y_{1:M}}^{-1} = \sum_{i=1}^{M} X_i^\top (\sigma_\theta^2 X_i X_i^\top + \sigma_1^2 I)^{-1} X_i, \tag{25}$$

$$\mu_{\tau|Y_{1:M}} = \Sigma_{\tau|Y_{1:M}} \sum_{i=1}^{M} X_i^\top (\sigma_\theta^2 X_i X_i^\top + \sigma_1^2 I)^{-1} \mathbf{y}_i \tag{26}$$

$$p(\boldsymbol{\theta}_{M+1}|Y_{1:M}, \mathbf{y}_{M+1}) = \mathcal{N}(\mu_{\boldsymbol{\theta}_{M+1}|Y_{1:M+1}}, \Sigma_{\boldsymbol{\theta}_{M+1}|Y_{1:M+1}}), \tag{27}$$

$$\Sigma_{\theta+\tau|Y_{1:M}} = \sigma_\theta^2 I + \Sigma_{\tau|Y_{1:M}} \tag{28}$$

$$\Sigma_{\boldsymbol{\theta}_{M+1}|Y_{1:M+1}}^{-1} = \sigma_{M+1}^{-2} X_{M+1}^\top X_{M+1} + \Sigma_{\theta+\tau|Y_{1:M}}^{-1} \tag{29}$$

$$\mu_{\boldsymbol{\theta}_{M+1}|Y_{1:M+1}} = \Sigma_{\boldsymbol{\theta}_{M+1}|Y_{1:M+1}} (\sigma_{M+1}^{-2} X_{M+1}^\top \mathbf{y}_{M+1} + \Sigma_{\theta+\tau|Y_{1:M}}^{-1} \mu_{\tau|Y_{1:M}}) \tag{30}$$

## C.3 Upper bound for meta linear regression

In this section we prove the main upper bound result of our paper, Theorem 4.

**Theorem 4** (Meta Linear Regression Upper Bound). *Let $\hat{\boldsymbol{\theta}}_{M+1}$ be the maximum-a-posteriori estimator, $\mu_{\boldsymbol{\theta}_{M+1}|Y_{1:M+1}}$. Then,*

$$R^*_{\mathcal{P}_{LR}} \leq \sup_{\boldsymbol{\theta}_1,\dots,\boldsymbol{\theta}_{M+1}\in\mathbb{B}_2(1)} \mathbb{E}[\|\hat{\boldsymbol{\theta}}_{M+1} - \boldsymbol{\theta}_{M+1}\|^2] \leq O\left(d\sigma^2_{M+1} C(M,n,k)^{-2} D(M,n,k)\right)$$

*where,*

$$C(M,n,k) = \left[k + \frac{Mn}{\frac{n(M+\kappa^2)s_2^2}{\alpha_2} + A}\right] \text{, and, } D(M,n,k) = \left[k + \frac{Mn}{(\frac{n}{L_1}+A_1)(\frac{Mn}{L_2}+A_2)}\right].$$

*Expectations are taken over the data conditioned on $\boldsymbol{\theta}_1,\dots,\boldsymbol{\theta}_{M+1}$. Additional terms not depending on $d$, $M$, $n$, $k$ are defined in Appendix C.2.*

Before proceeding with the proof, we introduce some additional notation and technical results.

**Additional notation** To alleviate (only a little of) the notational clutter, we will define the following quantities,

- $\Sigma' = \Sigma_{\boldsymbol{\theta}_{M+1}|Y_{1:M+1}}$
- $\Sigma'_0 = \Sigma_{\theta+\tau|Y_{1:M}}$.
- $s_{\min}(X/\sqrt{n}) = s_1$
- $s_{\min}(X_{M+1}/\sqrt{k}) = s_2$
- $s_{\max}(X/\sqrt{n}) = \gamma_1 = \kappa s_1$
- $s_{\max}(X_{M+1}/\sqrt{k}) = \gamma_2 = \kappa_{M+1}s_2$
- $\alpha_1 = \sigma_1^2/\sigma_\theta^2$
- $\alpha_2 = \sigma^2_{M+1}/\sigma_\theta^2$
- $L = \frac{\alpha_2}{(M+\kappa^2)s_2^2}$
- $L_1 = \frac{\alpha_1}{s_1^2 s_2^2 \kappa^2_{M+1}}$
- $L_2 = \frac{\tilde{\kappa}\kappa_\tau\alpha_2}{2s_2^2\kappa^2_{M+1}}$
- $A = \frac{s_2^2\alpha_1}{s_1^2\alpha_2}$
- $A_1 = s_2^2\kappa^2_{M+1}$
- $A_2 = \frac{\alpha_1 s_2^2 \kappa^2_{M+1}}{\kappa_\tau^2 s_1^2 \alpha_2}$

As we have uniform bounds on the singular values of all design matrices, we introduced an auxillary matrix $X$ whose largest and smallest singular values are given by $\sqrt{n}\gamma_1$ and $\sqrt{n}s_1$ respectively.

We will also write $S(A) = \text{Cov}[A, A]$, and $\kappa(A) = s_{\max}(A)/s_{\min}(a)$ throughout.

**Bias-Variance Decomposition** As is standard, we can decompose the risk into the bias and variance of the estimator:

$$\mathbb{E}[\|\hat{\boldsymbol{\theta}}_{M+1} - \boldsymbol{\theta}_{M+1}\|^2] = \mathbb{E}[\text{Tr}((\hat{\boldsymbol{\theta}}_{M+1} - \boldsymbol{\theta}_{M+1})(\hat{\boldsymbol{\theta}}_{M+1} - \boldsymbol{\theta}_{M+1})^\top)] \tag{31}$$

$$= \text{Tr}(\mathbb{E}[(\hat{\boldsymbol{\theta}}_{M+1} - \boldsymbol{\theta}_{M+1})(\hat{\boldsymbol{\theta}}_{M+1} - \boldsymbol{\theta}_{M+1})^\top]) \tag{32}$$

$$= \text{Tr}(\text{Cov}[\hat{\boldsymbol{\theta}}_{M+1}, \hat{\boldsymbol{\theta}}_{M+1}]) + \text{Tr}(\mathbb{E}[(\hat{\boldsymbol{\theta}}_{M+1} - \boldsymbol{\theta}_{M+1})]\mathbb{E}[(\hat{\boldsymbol{\theta}}_{M+1} - \boldsymbol{\theta}_{M+1})]^\top) \tag{33}$$

In the next two sections, we will derive upper bounds on the bias and variance terms above.

### C.3.1 VARIANCE TECHNICAL LEMMAS

We first decompose the variance into contributions from two sources: the variance from data in the novel task and the variance from data in the source tasks.

**Lemma 9.** *(Variance decomposition)* Let $\hat{\boldsymbol{\theta}}_{M+1} = \mu_{\boldsymbol{\theta}_{M+1}|Y_{1:M+1}}$ as defined above. Then the variance of the estimator can be written as

$$\text{Tr}(S(\hat{\boldsymbol{\theta}}_{M+1})) = \text{Tr}(\Sigma' \sigma_{M+1}^{-2} X_{M+1}^\top X_{M+1} \Sigma') + \text{Tr}(S(\Sigma' \Sigma_0'^{-1} \mu_{\tau|Y_{1:M}}))$$

*Proof.*

$$\text{Tr}(\text{Cov}[\hat{\boldsymbol{\theta}}_{M+1}, \hat{\boldsymbol{\theta}}_{M+1}]) = \text{Tr}(S(\Sigma'(\sigma_{M+1}^{-2} X_{M+1}^\top \mathbf{y}_{M+1} + \Sigma_0'^{-1} \mu_{\tau|Y_{1:M}}))) \tag{34}$$

$$= \text{Tr}(S(\Sigma' \sigma_{M+1}^{-2} X_{M+1}^\top \mathbf{y}_{M+1}) + S(\Sigma' \Sigma_0'^{-1} \mu_{\tau|Y_{1:M}})) \tag{35}$$

$$= \text{Tr}(\Sigma' \sigma_{M+1}^{-2} X_{M+1}^\top S(\mathbf{y}_{M+1}) \sigma_{M+1}^{-2} X_{M+1} \Sigma' + S(\Sigma' \Sigma_0'^{-1} \mu_{\tau|Y_{1:M}})) \tag{36}$$

$$= \text{Tr}(\Sigma' \sigma_{M+1}^{-2} X_{M+1}^\top X_{M+1} \Sigma') + \text{Tr}(S(\Sigma' \Sigma_0'^{-1} \mu_{\tau|Y_{1:M}})) \tag{37}$$

$\square$

We will now work towards a bound for each of the two variance terms in Lemma 9 separately. To do so, we will need to produce bounds on the singular values of terms appearing in Lemma 9.

We begin with the covariance term $\Sigma'$.

**Lemma 10.** *(Novel task covariance singular value bound)* Let $L$, $A$ and $s_2$ be as defined above. Then,

$$s_{\max}(\Sigma') \leq \frac{\sigma_{M+1}^2}{s_2^2} \left[ k + \frac{n}{\frac{Mn}{L} + A} \right]^{-1}.$$

*Proof.* Using Lemma 6, we can bound $s_{\max}(\Sigma')$ as follows,

$$s_{\max}(\Sigma') = s_{\max}(\Sigma_{\boldsymbol{\theta}_{M+1}|Y_{1:M+1}}) \tag{38}$$

$$= s_{\max}((\sigma_{M+1}^{-2} X_{M+1}^\top X_{M+1} + \Sigma_{\theta+\tau|Y_{1:M}}^{-1})^{-1}) \tag{39}$$

$$= 1/s_{\min}(\sigma_{M+1}^{-2} X_{M+1}^\top X_{M+1} + \Sigma_{\theta+\tau|Y_{1:M}}^{-1}) \tag{40}$$

$$\leq 1/(s_{\min}(\sigma_{M+1}^{-2} X_{M+1}^\top X_{M+1}) + s_{\min}(\Sigma_{\theta+\tau|Y_{1:M}}^{-1})) \tag{41}$$

$$= 1/(s_{\min}(\sigma_{M+1}^{-2} X_{M+1}^\top X_{M+1}) + 1/s_{\max}(\Sigma_{\theta+\tau|Y_{1:M}})) \tag{42}$$

Now, using the auxillary matrix $X$,

$$s_{\max}(\Sigma') \leq \left[ s_{\min}(\sigma_{M+1}^{-2} X_{M+1}^\top X_{M+1}) + \frac{1}{\sigma_\theta^2 + \frac{1}{M} s_{\max}((X^\top \tilde{C}^{-1} X)^{-1})} \right]^{-1} \tag{43}$$

$$= \left[ \frac{s_{\min}(X_{M+1}^\top X_{M+1})}{\sigma_{M+1}^2} + \frac{1}{\sigma_\theta^2 + \frac{1}{M} \frac{1}{s_{\min}(X^\top \tilde{C}^{-1} X)}} \right]^{-1} \tag{44}$$

$$\leq \left[ \frac{k s_{\min}^2(X_{M+1}/\sqrt{k})}{\sigma_{M+1}^2} + \frac{1}{\sigma_\theta^2 + \frac{1}{M} \frac{1}{s_{\min}(X^\top \tilde{C}^{-1} X)}} \right]^{-1} \tag{45}$$

$$\leq \left[ \frac{k s_2^2}{\sigma_{M+1}^2} + \frac{1}{\sigma_\theta^2 + \frac{1}{M} \frac{1}{s_{\min}(X^\top (X \sigma_\theta^2 I X^\top + \sigma^2 I)^{-1} X)}} \right]^{-1} \tag{46}$$

Above we have used Lemma 6 repeatedly, alongside the standard identity, $s_{\max}(A^{-1}) = s_{\min}(A)^{-1}$. We continue now, additionally using Lemma 7,

$$s_{\max}(\Sigma') \leq \left[ \frac{ks_2^2}{\sigma_{M+1}^2} + \frac{1}{\sigma_\theta^2 + \frac{1}{M} \frac{1}{s_{\min}(X^\top X) s_{\min}((X\sigma_\theta^2 I X^\top + \sigma_1^2 I)^{-1})}} \right]^{-1} \tag{47}$$

$$= \left[ \frac{ks_2^2}{\sigma_{M+1}^2} + \frac{1}{\sigma_\theta^2 + \frac{s_{\max}(X\sigma_\theta^2 I X^\top + \sigma_1^2 I)}{s_{\min}(X^\top X)}} \right]^{-1} \tag{48}$$

$$\leq \left[ \frac{ks_2^2}{\sigma_{M+1}^2} + \frac{1}{\sigma_\theta^2 + \frac{1}{M} \frac{s_{\max}(X\sigma_\theta^2 I X^\top) + \sigma_1^2}{ns_1^2}} \right]^{-1} \tag{49}$$

$$\leq \left[ \frac{ks_2^2}{\sigma_{M+1}^2} + \frac{1}{\sigma_\theta^2 + \frac{\sigma_\theta^2 s_{\max}(XX^\top) + \sigma_1^2}{ns_1^2}} \right]^{-1} \tag{50}$$

$$= \left[ \frac{ks_2^2}{\sigma_{M+1}^2} + \frac{1}{\sigma_\theta^2 + \frac{1}{M} \frac{\sigma_\theta^2 ns_1^2 \kappa^2 + \sigma_1^2}{ns_1^2}} \right]^{-1} \tag{51}$$

$$= \sigma_{M+1}^2 \left[ ks_2^2 + \frac{Mn}{\frac{n(M+\kappa^2)}{\alpha_2} + \frac{\alpha_1}{s_1^2 \alpha_2}} \right]^{-1} \tag{52}$$

$$= \frac{\sigma_{M+1}^2}{s_2^2} \left[ k + \frac{n}{\frac{Mn}{L} + A} \right]^{-1}. \tag{53}$$

$\square$

Next, we deal with terms appearing corresponding to the data from the source tasks.

**Lemma 11.** *(Source tasks covariance singular value bound)* Let $\tilde{C}_1 = \sigma_\theta^2 XX^\top + \sigma_1^2 I$, and write $\tilde{\kappa} = \kappa(\tilde{C}_1)$ and $\kappa_\tau = \kappa(\Sigma_{\tau|Y_{1:M}})$. Then,

$$s_{\max}{}^2(\Sigma_{\theta+\tau|Y_{1:M}}^{-1} \Sigma_{\tau|Y_{1:M}}) \leq \frac{1}{\frac{2M\sigma_\theta^2 ns_1^2}{s_{\max}(\tilde{C}_1)\kappa_\tau} + \frac{1}{\kappa_\tau^2}} =: D_1$$

*and,*

$$s_{\max}(\sigma_1^2 \tilde{C}_1^{-1}) \leq \frac{1}{\frac{ns_1^2}{\alpha_1} + 1} =: D_2$$

*Proof.* Using Lemma 6 and Lemma 7 we have,

$$s_{\max}{}^2(\Sigma_{\theta+\tau|Y_{1:M}}^{-1} \Sigma_{\tau|Y_{1:M}}) = s_{\max}{}^2(\Sigma_{\theta+\tau|Y_{1:M}}^{-1} \Sigma_{\tau|Y_{1:M}}) \tag{54}$$

$$\leq s_{\max}{}^2(\Sigma_{\theta+\tau|Y_{1:M}}^{-1}) s_{\max}{}^2(\Sigma_{\tau|Y_{1:M}}) \tag{55}$$

$$= s_{\min}{}^{-2}(\Sigma_{\theta+\tau|Y_{1:M}}) s_{\max}{}^2(\Sigma_{\tau|Y_{1:M}}) \tag{56}$$

$$= s_{\min}{}^{-2}(\sigma_\theta^2 I + \Sigma_{\tau|Y_{1:M}}) s_{\max}{}^2(\Sigma_{\tau|Y_{1:M}}) \tag{57}$$

$$\leq \frac{s_{\max}(\Sigma_{\tau|Y_{1:M}})^2}{(\sigma_\theta^2 + s_{\min}(\Sigma_{\tau|Y_{1:M}}))^2} \tag{58}$$

Now, using $\sigma_\theta^2 > 0$,

$$\frac{s_{\max}(\Sigma_{\tau|Y_{1:M}})^2}{(\sigma_\theta^2 + s_{\min}(\Sigma_{\tau|Y_{1:M}}))^2} \leq \frac{s_{\max}(\Sigma_{\tau|Y_{1:M}})^2}{2\sigma_\theta^2 s_{\min}(\Sigma_{\tau|Y_{1:M}}) + s_{\min}(\Sigma_{\tau|Y_{1:M}})^2} \tag{59}$$

$$\leq \frac{1}{\frac{2\sigma_\theta^2}{s_{\max}(\Sigma_{\tau|Y_{1:M}})\kappa_\tau} + \frac{1}{\kappa_\tau^2}} \tag{60}$$

Introducing the auxillary matrix $X$ and using Lemma 6 and Lemma 7 on $\Sigma_{\tau|Y_{1:M}}$, we have

$$\frac{1}{\frac{2\sigma_\theta^2}{s_{\max}(\Sigma_{\tau|Y_{1:M}})\kappa_\tau} + \frac{1}{\kappa_\tau^2}} \leq \frac{1}{\frac{2M\sigma_\theta^2 s_{\min}(X^\top \tilde{C}^{-1} X)}{\kappa_\tau} + \frac{1}{\kappa_\tau^2}}, \tag{61}$$

where,

$$s_{\min}(X^\top \tilde{C}_1^{-1} X) \geq \frac{s_{\min}(X^\top X)}{s_{\max}(\tilde{C}_1)} \tag{62}$$

$$= \frac{n s_1^2}{s_{\max}(\tilde{C}_1)}. \tag{63}$$

This gives the first stated inequality,

$$s_{\max}{}^2(\Sigma_{\theta+\tau|Y_{1:M}}^{-1} \Sigma_{\tau|Y_{1:M}}) \leq \frac{1}{\frac{2M\sigma_\theta^2 n s_1^2}{s_{\max}(\tilde{C}_1)\kappa_\tau} + \frac{1}{\kappa_\tau^2}} =: D_1 \tag{64}$$

The second follows as,

$$s_{\max}(\sigma_1^2 \tilde{C}_1^{-1}) = \frac{\sigma_1^2}{s_{\min}(\sigma_\theta^2 XX^\top + \sigma_1^2 I)} \tag{65}$$

$$\leq \frac{\sigma_1^2}{s_{\min}(\sigma_\theta^2 XX^\top) + s_{\min}(\sigma_1^2 I)} \tag{66}$$

$$\leq \frac{\sigma_1^2}{n s_1^2 \sigma_\theta^2 + \sigma_1^2} \tag{67}$$

$$= \frac{1}{\frac{n s_1^2}{\alpha_1} + 1} =: D_2 \tag{68}$$

$$\square$$

In Lemma 11, we introduced additional condition numbers, which we can bound as follows,

$$\tilde{\kappa} = \kappa(\tilde{C}_1) = \kappa(\sigma_\theta^2 XX^\top + \sigma_1^2 I) \leq \kappa(\sigma_\theta^2 XX^\top) \leq \kappa(\sigma_\theta^2 XX^\top)\kappa(\sigma_\theta^2 I) = \kappa^2, \tag{69}$$

$$\kappa_\tau = \kappa(\Sigma_{\tau|Y_{1:M}}) \leq \kappa(X^\top X)\kappa(\tilde{C}_1) = \kappa^2 \tilde{\kappa} \leq \kappa^4. \tag{70}$$

### C.3.2 VARIANCE UPPER BOUND

We are now ready to put the above technical results together to achieve a bound on the variance of the estimator.

**Lemma 12.** *(Variance bound)*

$$\mathrm{Tr}(S(\hat{\boldsymbol{\theta}}_{M+1})) \leq \frac{\kappa_{M+1}^2 \sigma_{M+1}^2}{s_2^2} d \left[ k + \frac{n}{\frac{n}{L} + A} \right]^{-2} \left[ k + \frac{Mn}{(\frac{n}{L_1} + A_1)(\frac{Mn}{L_2} + A_2)} \right]$$

*Proof.* First, by Lemma 9 we can decompose the overall variance into two terms:

$$\mathrm{Tr}(S(\hat{\boldsymbol{\theta}}_{M+1})) = \mathrm{Tr}(\Sigma' \sigma_{M+1}^{-2} X_{M+1}^\top X_{M+1} \Sigma') + \mathrm{Tr}(S(\Sigma' \Sigma_0'^{-1} \mu_{\tau|Y_{1:M}}))$$

We deal with the left term first.

Using trace permutation invariance and the von Neumann trace inequality (Lemma 8). We can upper bound the left variance term as follows,

$$\text{Tr}(\Sigma' \sigma_{M+1}^{-2} X_{M+1}^\top X_{M+1} \Sigma') = \sigma_{M+1}^{-2} \text{Tr}(\Sigma' \Sigma' X_{M+1}^\top X_{M+1}) \tag{71}$$

$$\leq dk \sigma_{M+1}^{-2} s_{\max}(\Sigma')^2 s_{\max}{}^2(X_{M+1}/\sqrt{k}) \tag{72}$$

$$= dk \sigma_{M+1}^{-2} s_{\max}(\Sigma')^2 s_2^2 \kappa_{M+1}^2 \tag{73}$$

For the second variance term, we observe that,

$$\text{Tr}(\Sigma' \Sigma_0'^{-1} S(\mu_{\tau|Y_{1:M}}) \Sigma_0'^{-1} \Sigma') \tag{74}$$

$$= \text{Tr}\left(\Sigma' \Sigma_0'^{-1} S\left(\Sigma_{\tau|Y_{1:M}} \sum_{i=1}^M X_i^\top (\sigma_\theta^2 X_i X_i^\top + \sigma_1^2 I)^{-1} \mathbf{y}_i\right) \Sigma_0'^{-1} \Sigma'\right) \tag{75}$$

$$\leq M \text{Tr}(\Sigma' \Sigma_0'^{-1} \Sigma_{\tau|Y_{1:M}} X^\top \tilde{C}^{-1} S(y_1) \tilde{C}^{-1} X \Sigma_{\tau|Y_{1:M}} \Sigma_0'^{-1} \Sigma') \tag{76}$$

$$= M \text{Tr}(\Sigma' \Sigma_0'^{-1} \Sigma_{\tau|Y_{1:M}} X^\top \tilde{C}_1^{-1} \sigma_1^2 I \tilde{C}_1^{-1} X \Sigma_{\tau|Y_{1:M}} \Sigma_0'^{-1} \Sigma') \tag{77}$$

$$\leq M s_{\max}(\Sigma')^2 s_{\max}{}^2(\Sigma_{\theta+\tau|Y_{1:M}}^{-1} \Sigma_{\tau|Y_{1:M}}) \sigma_1^2 \text{Tr}(X^\top \tilde{C}^{-1} \tilde{C}^{-1} X) \tag{78}$$

Using Lemma 11, we have,

$$\text{Tr}(\Sigma' \Sigma_0'^{-1} S(\mu_{\tau|Y_{1:M}}) \Sigma_0'^{-1} \Sigma') \leq s_{\max}(\Sigma')^2 M D_1 D_2 \text{Tr}(X^\top X) s_{\max}(\tilde{C}_1^{-1}) \tag{79}$$

$$\leq s_{\max}(\Sigma')^2 M D_1 D_2 \min(n, d) n s_{\max}(\tilde{C}_1^{-1}) \tag{80}$$

$$\leq s_{\max}(\Sigma')^2 D_2 \frac{M \min(n, d) n}{\frac{2M\sigma_\theta^2 n s_1^2 s_{\min}(\tilde{C}_1)}{s_{\max}(\tilde{C}_1)\kappa_\tau} + \frac{s_{\min}(\tilde{C}_1)}{\kappa_\tau^2}} \tag{81}$$

$$\leq s_{\max}(\Sigma')^2 D_2 \frac{M \min(n, d) n}{\frac{2M\sigma_\theta^2 n s_1^2 s_{\min}(\tilde{C}_1)}{s_{\max}(\tilde{C}_1)\kappa_\tau} + \frac{\sigma_1^2}{\kappa_\tau^2}} \tag{82}$$

$$\leq s_{\max}(\Sigma')^2 D_2 \frac{\min(n, d) n}{\sigma_{M+1}^2} \frac{M}{\frac{2Mn}{\tilde{\kappa}\kappa_\tau \alpha_2} + \frac{\alpha_1}{\kappa_\tau^2 \sigma_1^2 \alpha_2}} \tag{83}$$

$$\leq \frac{s_{\max}(\Sigma')^2}{\sigma_{M+1}^2} \frac{nd}{\frac{ns_1^2}{\alpha_1} + 1} \frac{M}{\frac{2Mn}{\tilde{\kappa}\kappa_\tau \alpha_2} + \frac{\alpha_1}{\kappa_\tau^2 s_1^2 \alpha_2}} \tag{84}$$

Finally, rearranging and using Lemma 10, we can bound the sum of the two terms in the variance as follows,

$$\text{Tr}(S(\hat{\boldsymbol{\theta}}_{M+1})) \leq \frac{s_{\max}(\Sigma')^2}{\sigma_{M+1}^2}\left(kds_2^2\kappa_{M+1}^2 + \frac{nd}{\frac{ns_1^2}{\alpha_1} + 1} \frac{M}{\frac{2Mn}{\tilde{\kappa}\kappa_\tau \alpha_2} + \frac{\alpha_1}{\kappa_\tau^2 s_1^2 \alpha_2}}\right) \tag{85}$$

$$\leq \frac{s_{\max}(\Sigma')^2 s_2^2 \kappa_{M+1}^2}{\sigma_{M+1}^2}\left(kd + \frac{nd}{\frac{ns_1^2 s_2^2 \kappa_{M+1}^2}{\alpha_1} + s_2^2 \kappa_{M+1}^2} \frac{M}{\frac{2Mns_2^2 \kappa_{M+1}^2}{\tilde{\kappa}\kappa_\tau \alpha_2} + \frac{\alpha_1 s_2^2 \kappa_{M+1}^2}{\kappa_\tau^2 s_1^2 \alpha_2}}\right) \tag{86}$$

$$\leq \frac{\kappa_{M+1}^2 \sigma_{M+1}^2}{s_2^2}\left[k + \frac{n}{\frac{n}{L} + A}\right]^{-2} \cdot \tag{87}$$

$$\left[kd + \frac{Mnd}{(\frac{ns_1^2 s_2^2 \kappa_{M+1}^2}{\alpha_1} + s_2^2 \kappa_{M+1}^2)(\frac{2Mns_2^2 \kappa_{M+1}^2}{\tilde{\kappa}\kappa_\tau \alpha_2} + \frac{\alpha_1 s_2^2 \kappa_{M+1}^2}{\kappa_\tau^2 s_1^2 \alpha_2})}\right] \tag{88}$$

$$\leq \frac{\kappa_{M+1}^2 \sigma_{M+1}^2}{s_2^2} d\left[k + \frac{n}{\frac{n}{L} + A}\right]^{-2}\left[k + \frac{Mn}{(\frac{n}{L_1} + A_1)(\frac{Mn}{L_2} + A_2)}\right] \tag{89}$$

$$\square$$

### C.3.3 BOUNDING THE BIAS

**Lemma 13.** *(**Bias upper bound**) Given $\theta_1, \ldots, \boldsymbol{\theta}_{M+1} \in \mathbb{B}_2(1)$, we have,*

$$\mathbb{E}[(\hat{\boldsymbol{\theta}}_{M+1} - \boldsymbol{\theta}_{M+1})] \leq O\left(d\left[k + \frac{Mn}{\frac{n(M+\kappa^2)s_2^2}{\alpha_2} + A}\right]^{-2}\right)$$

*Proof.* The bias can be computed as follows,

$$\mathbb{E}[(\hat{\boldsymbol{\theta}}_{M+1} - \boldsymbol{\theta}_{M+1})] = \mathbb{E}\mu_{\boldsymbol{\theta}_{M+1}|Y_{1:M+1}} - \boldsymbol{\theta}_{M+1} \tag{90}$$

$$= \Sigma_{\boldsymbol{\theta}_{M+1}|Y_{1:M+1}}\mathbb{E}(\sigma_{M+1}^{-2}X_{M+1}^\top y_2 + \Sigma_{\theta+\tau|Y_{1:M}}^{-1}\mu_{\tau|Y_{1:M}}) - \boldsymbol{\theta}_{M+1} \tag{91}$$

$$= \Sigma_{\boldsymbol{\theta}_{M+1}|Y_{1:M+1}}(\sigma_{M+1}^{-2}X_{M+1}^\top X_{M+1}\boldsymbol{\theta}_{M+1} + \Sigma_{\theta+\tau|Y_{1:M}}^{-1}\mathbb{E}\mu_{\tau|Y_{1:M}}) - \boldsymbol{\theta}_{M+1} \tag{92}$$

$$= (\sigma_{M+1}^{-2}X_{M+1}^\top X_{M+1} + \Sigma_{\theta+\tau|Y_{1:M}}^{-1})^{-1}\cdot \tag{93}$$

$$(\sigma_{M+1}^{-2}X_{M+1}^\top X_{M+1}\boldsymbol{\theta}_{M+1} + \Sigma_{\theta+\tau|Y_{1:M}}^{-1}\mathbb{E}\mu_{\tau|Y_{1:M}}) - \boldsymbol{\theta}_{M+1} \tag{94}$$

$$= (F + G)^{-1}(F\boldsymbol{\theta}_{M+1} + G\mu_{\tau|Y_{1:M}}) - \boldsymbol{\theta}_{M+1} \tag{95}$$

$$= (F + G)^{-1}F\boldsymbol{\theta}_{M+1} - (F + G)^{-1}(F + G)\boldsymbol{\theta}_{M+1} + (F + G)^{-1}G\mathbb{E}\mu_{\tau|Y_{1:M}} \tag{96}$$

$$= (F + G)^{-1}G(\mathbb{E}\mu_{\tau|Y_{1:M}} - \boldsymbol{\theta}_{M+1}), \tag{97}$$

where we wrote $F = \sigma_{M+1}^{-2}X_{M+1}^\top X_{M+1}$, and $G = \Sigma_{\theta+\tau|Y_{1:M}}^{-1}$. Thus,

$$\mathbb{E}[(\hat{\boldsymbol{\theta}}_{M+1} - \boldsymbol{\theta}_{M+1})]^\top \mathbb{E}[(\hat{\boldsymbol{\theta}}_{M+1} - \boldsymbol{\theta}_{M+1})] \leq \|(F + G)^{-1}\|_2^2\|G\|_2^2\|\mathbb{E}\mu_{\tau|Y_{1:M}} - \boldsymbol{\theta}_{M+1}\|_2^2$$

We can bound each term in turn. First, note that $\|G\|_2^2 \leq 1/\sigma_\theta^2$, and we have bounded $\|(F + G)^{-1}\|_2^2$ above. We can write,

$$\|\mathbb{E}\mu_{\tau|Y_{1:M}} - \boldsymbol{\theta}_{M+1}\|_2^2 = \left\|\Sigma_{\tau|Y_{1:M}}\left(\sum_{i=1}^{M}X_i^\top \tilde{C}_i^{-1}X_i\theta_i\right) - \boldsymbol{\theta}_{M+1}\right\|_2^2$$

$$\|\mathbb{E}\mu_{\tau|Y_{1:M}} - \boldsymbol{\theta}_{M+1}\|_2^2 = \left\|\Sigma_{\tau|Y_{1:M}}\left(\sum_{i=1}^{M}X_i^\top \tilde{C}_i^{-1}X_i\theta_i\right) - \Sigma_{\tau|Y_{1:M}}\Sigma_{\tau|Y_{1:M}}^{-1}\boldsymbol{\theta}_{M+1}\right\|_2^2$$

$$= \left\|\Sigma_{\tau|Y_{1:M}}\sum_{i=1}^{M}X_i^\top \tilde{C}_i^{-1}X_i(\theta_i - \boldsymbol{\theta}_{M+1})\right\|_2^2$$

$$\leq \left(\sum_{i=1}^{M}\|\Sigma_{\tau|Y_{1:M}}\|_2 \|X_i^\top \tilde{C}_i^{-1}X_i(\theta_i - \boldsymbol{\theta}_{M+1})\|_2\right)^2$$

$$\leq \left(\sum_{i=1}^{M}\|\Sigma_{\tau|Y_{1:M}}\|_2 \|X_i^\top \tilde{C}_i^{-1}X_i\|_2\right)^2$$

The last line follows from the fact that the parameters lie in a ball of unit radius. We now proceed by bounding the sum by $M$ times the supremum — with some light abuse of notation,

$$\|\mu_{\tau|Y_{1:M}} - \boldsymbol{\theta}_{M+1}\|_2^2 \leq (s_{\max}(X^\top \tilde{C}^{-1}X)s_{\max}(X\tilde{C}^{-1}X))^2$$

$$= s_{\max}(X^\top \tilde{C}^{-1}X)^4 \leq O(1)$$

Thus, overall the convergence of the bias is bounded by,

$$\mathbb{E}[(\hat{\boldsymbol{\theta}}_{M+1} - \boldsymbol{\theta}_{M+1})]^\top \mathbb{E}[(\hat{\boldsymbol{\theta}}_{M+1} - \boldsymbol{\theta}_{M+1})] \leq O(s_{\max}(\Sigma')^2) \leq O\left(d\left[k + \frac{Mn}{\frac{n(M+\kappa^2)s_2^2}{\alpha_2} + A}\right]^{-2}\right)$$

□

The proof of Theorem 4 is given by the combination of Lemma 12 and Lemma 13, and the bias-variance decomposition of the risk .

# D   ADDITIONAL EXPERIMENT DETAILS

## D.1   HIERARCHICAL BAYES EVALUATION

We sample $M$ linear models according to the hierarchical model in Section 5, with design matrices constructed by uniformly sampling points, $x \sim U[-1, 1]$, and storing the vector $\mathbf{x}_j = x^j$, for $i = 0, \ldots, d$ in each row of $X_i$.

To produce the plots in Figure 2 we computed the average loss over 100 random draws of the training data and labels from the same set of fixed $\boldsymbol{\theta}_{1:M+1}$ values. The $\boldsymbol{\theta}$ values were sampled once from the hierarchical model with $\tau = [0, 1, 2, 0, 0, 3, 1]$, and $\sigma_\theta^2 = 0.1$

Code to reproduce these plots is provided in the supplementary materials with our submission.

## D.2   SINUSOID REGRESSION WITH MAML

| Hyper parameters | Description |
|---|---|
| $\sigma$ | noise at test time. |
| M | number of tasks at the training tasks |
| $M_q$ | number of tasks at the testing tasks |
| eps_per_batch | episode per batch |
| train_ampl_range | range of amplitude at training |
| train_phase_range | range of phase at training |
| val_ampl_range | range of amplitude at testing |
| val_phase_range | range of phase at testing |
| inner_steps | number of steps of Maml |
| inner_lr | learning rate used to optimize parameter of the model |
| meta_lr | used to optimize parameter of the meta-learner |
| n | number of datapoints at training tasks(support set) |
| k | number of datapoints at testing tasks (support set) |
| $n_q$ | number of datapoints at training tasks (query set) . |
| $k_q$ | number of datapoints at testing tasks (query set). |

For all of these experiments we used a fully connected network with 6 layers and 40 hidden units per layer. The network is trained using the MAML algorithm (Finn et al., 2017) with 5 inner steps using SGD with an inner learning rate of $10^{-3}$. We used Adam for the outer loop learning with a learning rate of $10^{-3}$.

Expected error was computed after 500 epochs of optimization and was averaged over 30 runs. We produced our results through a comprehensive grid search over 72 combinations of the settings below and it required around 30 minutes to produce the output of each setting, using a system with 1 gpu and 3 cpus. This experiment therefore lasted 20 hours in total.
$M = 50, n \in \{20, 200\}, k \in \{100, 1000\}, \sigma \in [10^{-8}, 1.5], M_q = 100$, eps_per_batch $= 25$, train_ampl_range $= [1, 4]$, train_phase_range $= [0, \pi/2]$, val_ampl_range $= [3, 5]$, val_phase_range $= [0, \pi/2]$, inner_steps $= 5$, inner_lr $= 10^{-3}$, meta_lr $= 10^{-3}$

