# OpenReview forum: "Theoretical bounds on estimation error for meta-learning"
_ICLR.cc/2021/Conference — ICLR 2021 Poster_

### Official Review · AnonReviewer2 · 2020-10-14
**A good paper that investigate the fundamental limits of meta-learning in the sense of minimax novel-task risk**

**Rating:** 7
**Confidence:** 4

**Review:**

This paper provides a minimax novel-task risk lower bound for meta learning via information-theoretical techniques, showing the fundamental limits of meta learning. The novel-task minimax risk depends on the number of samples from the meta-training set and novel task, as well as the task similarity. The authors further investigate the meta learning problem on a hierarchical Bayesian model, discuss the lower bound and upper bound with maximum-a-posterior estimator.

Overall I feel this paper study an important problem of the meta-learning and the derivation all looks correct. The main drawback is that the presented minimax lower bound is quite pessimistic that may not fully exploit the potential task similarity in practice, but I feel it is still acceptable to first study the minimax lower bound, and leave the case with more structure as future work.

I would like to ask if there is a gap between the description at the beginning of Section 5 and the analysis starting from Section 5.1, as the bayesian model described at the beginning of Section 5 does not assume theta have ell_2 norm smaller or equal than 1, and in fact, cannot be universally hold as theta is sampled from a d dimensional Gaussian (can have high probability arguments instead). However, the analysis throughout Section 5 all have this constraint. I don’t see strong dependency on the norm of theta (e.g. in the covering number, KL bound etc) but I hope the authors check and clarify it in the next version.

One small tip: lower bound is better represented via big-Omega notation.

---

> ### Author Response · Authors · 2020-11-15
> **Thank you for your feedback!**
>
> ### Gap between 5 and 5.1
>
> We acknowledge that this is a potential source of confusion in the current write-up, and will clarify it in the next version.
>
> Here we present an alternative view-point. We have a meta-learner in Section 5 defined by Eq 28-30, that utilizes data from the training tasks and novel task. This meta-learner is derived by taking an empirical Bayes estimate in the hierarchical Gaussian model. The bounds we derive apply in the minimax setting that we study in Sections 3 and 4, even if the model is mis-specified for the actual data distribution seen. Natural extensions of our results may introduce high-probability arguments as you suggested, to bridge the gap.
>
> ### No strong dependency on the norm of $\theta$
>
> We assume that $\theta$ is bounded in a 1-norm ball. The norm of $\theta$ plays a role in the computation of the KL-divergence bounds in the lower-bounds, and in the bias computation in the upper-bounds. But due to the unit-factor this isn't visible in the presented results themselves.

---

> > ### Comment · AnonReviewer2 · 2020-11-15
> > **Thanks for your response. Can you make a revision based on the review and the discussion with Ahmad?**
> >
> > Thanks for your response. My questions in Section 5 are almost solved. On the other hands, can you make a revision based on the review and discussion with Ahmad? At the first glance I feel the results are reasonable and don't go into the details, but some of the suggestions from Ahmad's are valid, and I hope the authors can make some revision based on that, which will also improve the paper's quality.

---

> > > ### Author Response · Authors · 2020-11-16
> > > **We agree, and will be including changes in revision**
> > >
> > > Thank you for the timely reply.
> > >
> > > We agree that Ahmad has raised several valuable points, and we will be working on these in our revised version (hopefully completed within the next few days).

---

### Official Review · AnonReviewer1 · 2020-10-28

**Rating:** 7
**Confidence:** 4

**Review:**

The paper studies the information-theoretic lower bounds in the minimax setting of meta-learning. The paper also discusses upper and lower bounds in the hierarchical Bayesian framework of meta linear regression. The novelty of the paper is two-fold: a) it proves a novel meta-learning local packing result to compute the conditional information between training task samples and the novel task data distribution and b) it compares the lower bound of the risk to the risk of posterior estimate in meta linear regression. In addition, the authors verify the dependence of risk on various parameters in 2 different experiments.

My major concern is that in theorem 3, $Mn$ needs to grow exponentially with $d$ to be significantly better than the case when there are no training tasks available (theorem 2). However, theorem 4 states that $Mn$ just needs to depend polynomially on the dimension $d$. Hence, the upper bound and the lower bound on the risk have a gap on the parameter $d$. Thus, it is not clear if the lower bound in Theorem 1 is tight. It would be great to have a discussion on this and the assumptions that one needs to take to improve upon this gap.

Few other concerns:
1. In the hierarchical regression setting, the assumptions for theorem 3 and theorem 4 are different. A discussion on how the lower bounds will change, if we assume different variance of task parameters $\sigma_{\theta_1}, \cdots, \sigma_{\theta_{M+1}}$ and task-specific observed samples $\sigma_1, \cdots, \sigma_{M+1}$, will be helpful.
2.  The proof of theorem 3 focuses on packing the parameter space $\theta$ by an $\epsilon$-net on $B_2(4\delta)$, where $\delta$ is fixed later on. However, the model assumes $\theta$ coming from ball $B_2(1)$. There seems to be some mismatch in the theorem statement and the proof.
3. The risk in theorem 4 involves few more parameters whose effect on the risk hasn't been checked in experiments e.g. the ratio of variance in observed samples of novel task v/s variance in task parameters. Also, it will be interesting to see how the results change when the data distribution is changed since the risk depends on the singular values of the data matrix.

I have read the proofs. They are easy to read and understand. My scores are slightly on the lower side because I am concerned about the tightness of the lower bound in Theorem 1. I am happy to discuss this with the authors and other reviewers during the discussion period.

**After Rebuttal**
I have read the reviews of other reviewers and the responses of the authors to the questions posed by the reviewers and Ahmad Beirami. I understand that the authors have taken a pessimistic approach to compute the lower bounds of risk in the minimax setup of meta-learning. However, I believe the paper is an important step in this direction. Theorem 3 and Theorem 4 are important additions to the paper, which show a margin between the pessimistic lower bound of the risk and the actual risk with the introduction of structure to the learning setup. Overall,  I enjoyed reading the paper, and I have increased my score after the rebuttal.

---

> ### Author Response · Authors · 2020-11-15
> **Thank you for your feedback!**
>
> ### Tightness of Theorem 3 vs Theorem 4
>
> There is indeed a gap in the derived lower and upper bounds. We discuss this gap in Section 5.3 but will provide some additional discussion. We expect that our upper-bounds are asymptotically tight, and that the looseness is due to the relatedness assumptions used in Theorem 1, which are likely too weak for meta-learning in the hierarchical Gaussian model. We also discuss the inverse exponential scaling w.r.t dimension $d$ at the end of Section 5.1. We aim to (partially) address this gap through Corollary 2, which provides tighter lower bounds in the setting where the environment is only partially observed.
>
> ### Different variance assumptions
>
> Theorem 4 has a more general variance assumption (allowing different variances for the training tasks and novel task). We could introduce more general variance assumptions in Theorem~3, and need only derive a new upper-bound on the pairwise KL-divergences.
>
> ### Smaller bounding ball in proof
>
> This is a standard technique that is typically hidden in the proof of these minimax results (see e.g. Raskutti et al. 2011). While this step looks to be suboptimal, existing minimax results using this approach have provably optimal rates. Introducing a larger packing set makes the derivation much less clean; however, we are exploring this approach to see if this improves the bounds.
>
> ### Parameters in empirical evaluation
>
> Thank you for the suggestion. The settings we explored probe at the prominent model parameters that appear in our bound. However, we did not explore varying the data distribution. We will explore this and hope to include these results before the end of the rebuttal period.

---

### Official Review · AnonReviewer3 · 2020-10-28
**Good results; Suggestions for improving presentation**

**Rating:** 6
**Confidence:** 3

**Review:**

The authors study the performance of a meta-learner theoretically in two settings. In the first one the overall number of possible tasks is limited and tasks are close in KL-divergence. The second setting is MAP estimation for a family of linear regression tasks. Lower and upper bounds are provided on minimax parameter estimation error.

The bounds solidify common sense knowledge about the role of the number of training tasks, number of samples, and task relatedness and diversity on the performance of meta-learner and are supported by numerical examples. I am in favor of accepting this paper. I'll provide suggestions for improving presentation.

1. The assumption in the first setting is that the tasks are close in distribution space but far away in parameter space. Could you mention some examples of such set of tasks? It does not seem like a weak assumption to me, as mentioned in the paper, but then this is the nature of lower bounds. The point is not to model a typical scenario but to design hard cases that lead to high error. But in any way an example is necessary to show that this situation is possible.

2. It seems to me that the lower bound in the first setting has less to do with the performance of the meta-learner than suitability of parameter estimation error as a measure of performance. At the end of the day, it is accurate predictions, and not accurate parameters, that matter. The setting is designed so that tasks have similar distributions but quite different optimal parameters, which results in a high parameter estimation error for the meta-learner. It could be that the meta-learner in fact makes quite accurate predictions (since the tasks are actually close in distribution space) despite its high parameter error.

3. The first setting is worded generally (novel task generalization) while the second setting is presented as a more specific scenario (hierarchical Bayesian linear regression). The first time I was reading the paper I assumed that the first part describes general results on meta-learning and the second part shows a particular instance of the previous section, which was confusing as the second setting does not satisfy the assumptions in the first setting. I suggest presenting these two sections as two particular and different settings as they are.

Minor comments:

1. The statement of Theorem 1 has undefined notation. I assume I(.;.) is mutual information but it is not defined. The text defines W|\pi and Z|\pi but the equation uses \pi_{M+1} and W and Z separately. What do \pi_{M+1}, W, and Z (without the vertical bar) mean?

2. The paragraph above Corollary 1 is hard to understand. Do "training task" and "meta-training task" have the same meaning? Also, if the number of previous tasks is M \leq J-1, how can there be J previous tasks that are close?

3. What is B_2(1) in 5.1?

---

> ### Author Response · Authors · 2020-11-15
> **Thank you for your feedback!**
>
> ### Local-packing examples
>
> Our paper includes such a set of tasks in the space of linear regression models that we consider (Section 5.1).
>
> In general, it is difficult to construct realistic spaces of distributions that satisfy these constraints. Fortunately, by design, these assumptions align with those used in the standard minimax framework. Therefore, we may use our results to study meta-learners in spaces of distributions studied in those works.
>
> For example, Raskutti et al. provide several compelling examples, focused on $\ell_q$ high-dimensional linear regression. Yang \& Barron analyze local-packing for minimax rates, and provide several examples including data compression and non-parametric regression.
>
> ### Parameter estimation vs. accuracy
>
> We follow the typical statistical minimax risk framework where parameter estimation error is the focus. You bring up a good point, that these parametric estimation bounds do not speak to prediction performance in general. However, there are standard methods to extend the symmetric parameter loss to objectives like cross entropy. The clearest example we’re aware of is given in Duchi’s 2016 lecture notes, in Question 7.5. In short, the $\delta$-packing can be applied in loss space directly to allow more general forms of risk.
>
> ### Discrepancy between Section 4 and Section 5
>
> Our bounds in Section 5 are valid bounds on the minimax risk object that we introduce in Section 3. We agree that there are some discrepancies here that can be clarified significantly (see response to R2, "Gap between 5 and 5.1"). We are not certain what particular assumptions you are referring to, and so would appreciate clarification if necessary.
>
> #### Minor comments
>
> 1. Thank you, we will make these clear in the paper.
> 2. Yes, these have the same meaning. We will make these consistent. $M$ tasks are visible to the learner, but the proof requires $J$ distributions in the local-packing. In the setting $M < J-1$, we assume that the learner only sees tasks from some incomplete subset of the full packing.
> 3. This is a ball of radius 1 (measured in 2-norm), centered at the origin.

---

### Official Review · AnonReviewer4 · 2020-10-28
**Well written paper with potentially interesting insights on the theory of meta learning but some issues exist**

**Rating:** 5
**Confidence:** 4

**Review:**

Pros:
1 - The paper seems to be well written, have a good review of the references and necessaries for understanding the problem.
2 - Some of the results found in the paper seem to be interesting. For instance, the asymptotic analysis provided in paper gives some insight on the performance of meta learning algorithms with the number shots, ratio of observation noise to the sampling noise and the number of tasks.

Cons:
1 - It seems that most of the results of the paper are based on Loh (2017). It is expected that the author differentiate their contributions  with those of that reference more clearly.
2 - Some of the notations are misleading. This is a minor issue and up to the authors to change it or not but it may help with the readability of the paper. Some notations like $\theta$ are usually used for parameters in models. In this paper, $\theta$ is used as a function. I understand why the authors decided to use it as a function but it may be a bit misleading.
3 - Some of the assumptions in the paper can be very restrictive. For instance, it is assumed that the distributions are $2\delta$-separated while close in the KL divergence sense. Isn't this too restrictive? Maybe it is good for the authors to try to talk more about the implications of such assumptions. How does this restrict the space of interested probability distributions?
4 - Another example of a restrictive assumption is bounded minimum singular values. How does such a restriction affect the space of considered solutions?
5 - There has been no effort in comparing with any bounds that are available currently.

---

> ### Author Response · Authors · 2020-11-15
> **Thank you for your feedback!**
>
> ### Contribution relative to Loh 2017
> We certainly utilize some of the techniques presented in Loh 2017, and elsewhere in the minimax risk literature. We extend these results on several fronts: addressing generalization to novel tasks, multiple sources of training data, and novel local-packing bounds under product and mixture sampling procedures. We will make these contributions clearer in our paper.
>
> ### $\theta$ as a functional
>
> We follow the notation introduced by Loh, 2017 and others within the minimax risk literature. We acknowledge that this notation does not agree with the standard notation used in machine learning and related fields. But it does have the advantage of being succinct.
>
> We introduce short-hand notation ($\theta_P = \theta(P)$), which aligns with the standard ML notation.
>
> ### Local-packing is a strong constraint
> This is indeed a strong constraint, but is a typical one in the minimax risk setting. Further, these constraints are natural and relevant in FSL. For example, metric learning approaches to FSL such as Prototypical Networks learn an embedding space in which classes of images must be discriminable (meaning separated in the embedding space), but share features, so that novel class data are embedded within the bounds of the space.
>
> Notably, in this paper we do not introduce any additional constraints relative to the standard minimax setting. Therefore, existing packing sets can be utilized in our setting. Additionally, please see our response to R3 ("Local-packing examples").
>
> ### Minimum singular values assumption
>
> This assumption is potentially strong, but can be verified for large random matrices (Raskutti et. al, 2011). Note that our upper bound depends explicitly on the singular values lower-bound, and so the effect of weakening this assumption on our results can be measured.
>
> ### No comparison to existing bounds
>
> This is true, though we argue that our bounds are not immediately comparable to existing ones as, to our knowledge, these are the first minimax bounds in this meta-learning setting. We can directly compare our bounds to existing IID bounds in the minimax literature (and do so within our work). If you have any particular bounds in mind, we would be happy to consider how they compare to our results.

---

### Public Comment · ~Ahmad_Beirami1 · 2020-11-12
**minimax risk with side information is equal to that with no side information**

The authors define a notion of **minimax risk with side information** in (2), which is denoted by $R^\star_{\cal P}$.  Unfortunately, it appears that $R^\star_{\cal P} = R^\star$, i.e.,  with the way the authors define risk, side information does not help in the worst-case, which also intuitively makes sense because in the worst case data points from other tasks can be arbitrarily unrelated to the task at hand! Hence, all of the lower bounds that are developed by authors on $R^\star_{\cal P}$ are also lower bounds on the original minimax risk $R^\star$ in (1). From a theoretical standpoint, the authors' finding that the value of the new lower bounds is smaller than those of the original ones for the minimax risk is expected because *the authors are deriving looser lower bounds on the same minimax risk* $R^\star$.

**Why is $R^\star_{\cal P} = R^\star$?**
Let $R = E_{P_1, \ldots, P_{M+1}} \ell (\hat{\theta}, \theta_{P_{M+1}})$. The authors are defining the following quantity as the minimax risk: $R^\star_{\cal P} = \inf_{\hat{\theta}} \sup_{P_1, ..., P_{M+1}} R.$ It is straightforward to argue that $R^\star_{\cal P} \leq  R^\star$. In the following we also argue that $R^\star_{\cal P} \geq  R^\star,$ completing the argument.

We can bound

$R^\star_{\cal P} = \inf_{\hat{\theta}} \sup_{P_1, ..., P_{M+1}} R \geq \sup_{f(P_1, ..., P_{M+1})} \inf_{\hat{\theta}} E_f R$,

where $\sup$ is over all distributions $f$ supported on $\cal P$. We can further lower bound the RHS quantity by assuming that $P_1,..., P_{M+1}$ are chosen i.i.d.:

$\sup_{f(P_1, ..., P_{M+1})} \inf_{\hat{\theta}} E_f R \geq \sup_{f_1(P_1), ..., f_{M+1}(P_{M+1})} \inf_{\hat{\theta}} E_{f_1}...E_{f_{M+1}} R$.

Now, $R$ is constant with respect to $P_1, \ldots, P_M$ ($\hat{\theta}$ would not depend on $P_1, \ldots , P_M$ which are independent of $P_{M+1}$), and hence taking the expectations we have:

$\sup_{f_1(P_1), \ldots, f_{M+1}(P_{M+1})} \inf_{\hat{\theta}} E_{f_1} \ldots E_{f_{M+1}} R = \sup_{f_{M+1}(P_{M+1})} \inf_{\hat{\theta}} E_{f_{M+1}} R$.

If regularity conditions hold for the minimax risk to satisfy a minimax theorem (such as Sion's min-max theorem), we would further have:

$\sup_{f_{M+1}(P_{M+1})} \inf_{\hat{\theta}} E_{f_M+1} R = \inf_{\hat{\theta}} \sup_{P_{M+1}} R$.

The RHS is the single-task minimax risk, which would then imply that $R^\star_{\cal P}\geq R^\star$. Together with the fact that $R^\star_{\cal P} \leq  R^\star$, we conclude that $R^\star_{\cal P} =  R^\star$.


**Where is the problem?** The problem arises from the fact that there is nothing in the authors' definition to ensure that the knowledge of $P_1, \ldots, P_{M}$ (let alone samples from them) would restrict the possibilities for $P_{M+1} \in {\cal P}$. Even with perfect knowledge of $P_1, \ldots, P_{M}$, nature can choose $P_{M+1}$ arbitrarily on the entire $\cal P$, and hence the minimax risk would remain unchanged from the single-task case.

**How can this be resolved?** There are two ways to fix it:
1) In min-max world: Enforce some relatedness between $P_{M+1}$ and $P_1,\ldots, P_M$ via an additional constraint, perhaps by considering them to be close in some distance/divergence metric when the authors define the $\inf$.
2) In max-min world: the authors could also alternatively go Bayesian and define max-min risk and enforce some dependence on the structure of the distributions they define the $\sup$ over (to avoid the possibility that task M+1 be chosen independently from other tasks), which is similar to what they do in the example of Sec 3.

---

> ### Author Response · Authors · 2020-11-13
> **Side information matters with assumptions on the space of distributions**
>
> Hi Ahmad,
>
> Thank you for your detailed comments regarding our technical contributions. We’re pleased to have the opportunity to discuss this further. We will first provide a high-level discussion, explaining how our contributions are relevant in light of your comment.
>
> At a high level we agree with one direction of your argument, but it does not capture the full scope of Theorem 1, and our other results. The main contribution of our theorems is to specify how the sample complexity/error varies as a function of the relatedness of the tasks/distributions in ${\mathcal P}$.
>
> > "The authors define a notion of minimax risk with side information in (2), which is denoted by $R_{\mathcal{P}}^*$. Unfortunately, it appears that , $R^* = R_{\mathcal{P}}^*$,  i.e., with the way the authors define risk, side information does not help in the worst-case, which also intuitively makes sense because in the worst case data points from other tasks can be arbitrarily unrelated to the task at hand! "
>
>
> We agree with this statement partially, assuming that what you mean by worst-case is wrt the choice of ${\mathcal P}$. As your example shows, when the tasks in ${\mathcal P}$ are unrelated to one another, in the meta-learning setup (where $n$ samples of each of $M$ tasks $P_1,\ldots,P_M$ are given followed by $k$ samples from the $M+1$st=target task), the risk $R_{\mathcal{P}}^*$ will be the same as the baseline/iid risk, $R^*$ (where we just see $k$ samples from the target task).
>
> > "Hence, all of the lower bounds that are developed by authors on $R_{\mathcal{P}}^*$ are also lower bounds on the original minimax risk."
>
> Our lower bound extends the original/iid minimax risk. It is true that in the case where the tasks in $\mathcal{P}$ are unrelated our lower bound matches the iid case, as expected. However, our formulation also covers cases where the tasks are related to one another. An extreme example is the counter-point of the above example, where all distributions in ${\mathcal P}$ are very related: the meta-learning risk $R_{\mathcal{P}}^*$ will clearly be much lower than the iid risk $R^*$. Both the upper bounds and the lower bounds show this tradeoff. In such cases our lower bound differs from the iid situation. The mutual information term in the bound in Theorem 1 exactly captures the relationship between the tasks.
>
> You suggest one way to resolve this: "In min-max world: Enforce some relatedness between $P_{M+1}$ and $P_1,\ldots,P_M$ via an additional constraint, perhaps by considering them to be close in some distance/divergence metric when the authors  define the inf."
>
> This is exactly what we do in Corollary 1, where we constrain pairs of tasks in $\mathcal{P}$ to have a KL-divergence less than $\alpha$. This implies a high MI, which is why the risk decreases. In this case we clearly see the benefit of these $Mn$ samples from the $P_1,\ldots,P_M$ tasks in the first stage of meta-learning, as the worst-case risk is lowered by a term that depends on $Mn$.
>
> To conclude, consider an explicit counter-example where $R_{\mathcal{P}}^* \neq R^*$. Let $\mathcal{P} = \\{P_1, P_2, P_3\\}$, with $P_i = \textrm{Ber}(p_i)$, such that $|p_i - p_j| < \\epsilon \ll 1$ for all $i \neq j$. Let $M=2$, $n$ be large, and $k$ small. Now consider the naive meta-learner that treats all data as iid and computes the MLE. This estimator will certainly outperform the minimax-optimal estimator in the iid setting that uses only $k$ samples. This case where the distributions are strongly related serves as a simple counter-example to your proof.
>
> We will revise the paper to make this main point clearer: the meta-learning lower bound **is** weaker than the iid LB in some cases -- when the tasks in ${\mathcal P}$ are related (as quantified by mutual information/KL divergence). This is natural, and will match the upper bound.
>
> P.S. We are actively working on responses to our official reviewers and are updating our paper following their feedback. Thank you all for your patience!

---

> > ### Public Comment · ~Ahmad_Beirami1 · 2020-11-13
> > **Task-relatedness cannot be supported in the minimax framework with the way currently Section 3 is set up.**
> >
> > I appreciate the quick response of the authors.
> >
> > Let me explain more clearly where I believe the issue lies. With your notation (As you define in Section 3 and also in Appendix A), $\cal P$ is a set of distributions supported on $\cal Z$. In (2), you are taking a sup over $P_1, \ldots, P_{M+1} \in {\cal P}.$ This is mathematically equivalent to taking a sup over $(P_1, \ldots, P_{M+1}) \in {\cal P}^{M+1}$. This definition mathematically does not support any structure (task-relatedness constraint) among the meta-tasks that you are interested in.
> >
> > Regarding your counter-example, with $\cal P =${$P_1, P_2, P_3$} only allowing three values. In this example, in fact $R^\star = R^\star_{\cal P}$. Your novel task can be any member of the set {$P_1, P_2, P_3$} a priori for the calculation of $R^\star$. Given you observe $M$ tasks as members of {$P_1, P_2, P_3$}, a novel $M+1$-th task is still going to be a member of $\cal P =${$P_1, P_2, P_3$}, and $R^\star = R^\star_{\cal P}$. You may have meant for $\cal P$ to be the set of all Bernoulli sources with parameter $p \in (0,1),$ in which case there would be no room in this definition to enforce the desired task-relatedness $|p_i - p_j|<\epsilon$ within this framework.
> >
> > To remedy this, you could define your minimax risk by taking the sup over a set like
> > ${\cal Q}^{M+1}_\epsilon =$ {$ (P_1, \ldots, P_{M+1}) \in {\cal P}^{M+1}|  D(P_i||P_j) \leq \epsilon \quad \forall i,j \in [M+1]$ }
> > to enforce task-relatedness. Such structure is what you implicitly assume in building your example but the mathematical framework that you have laid down in Section 3 does not support this example.
> >
> > Finally, I agree with you that Corollary 1 still applies if you set up the problem in Section 3 to enforce a task-relatedness constraint in the minimax setup.

---

> > > ### Author Response · Authors · 2020-11-13
> > > **Revisiting our counter-example**
> > >
> > > > "This definition mathematically does not support any structure (task-relatedness constraint) among the meta-tasks that you are interested in."
> > >
> > > This definition does not impose any structure, that comes later in Section 4. But importantly, this definition does not prevent structure from being introduced.
> > >
> > > We disagree with your evaluation of our counter-example. We agree that all tasks are members of our three-distribution set $\mathcal{P}$, but this does not imply that $R^* = R^*_{\mathcal{P}}$.
> > >
> > > We'll expand on this briefly. We take our meta-learner as $\hat{\theta} = \frac{1}{Mn + k} (\sum_{z \in S_{1:M}} z + \sum_{z' \in S_{M+1}} z')$. Clearly this estimator will have a lower error-rate than $\frac{1}{k}\sum_{z' \in S_{M+1}} z'$ (remember that $k$ is small). In fact, as $n \rightarrow \infty$, the error will be $O(\epsilon)$ for the meta-learner (upper bounding $R_{\mathcal{P}}^*$) and something like $O(1/k)$ for the iid learner. Clearly, $R_{\mathcal{P}}^* < R^*$.
> > >
> > > > "You may have meant for $\mathcal{P}$ to be the set of all Bernoulli sources with parameter $p \in (0,1)$"
> > >
> > > No, we explicitly constructed $\mathcal{P}$ to contain only a small number of highly-related distributions. This is permitted in our framework of Section 3.
> > >
> > > > "To remedy this, you could define your minimax risk by taking the sup over a set like $\mathcal{Q}^{M+1}_{\epsilon} = \\{(P_1, \ldots, P_{M+1}) \in \mathcal{P}^{M+1}\vert D(P_i||P_j) \leq \epsilon\\}$"
> > >
> > > We do use a constraint in Section 4 which is, in essence, similar to this. In any case, Section 3 permits us to use $\mathcal{Q}^{M+1}_{\epsilon}$ in place of $\mathcal{P}^{M+1}$ --- this is not a problem. We feel that you are assuming that $\mathcal{P}$ must necessarily be the space of all possible distributions: this is what we meant by "worst case with respect to $\mathcal{P}$". But this is **not** the case. Our definition makes no strong assumptions about $\mathcal{P}$, and certainly supports spaces of distributions with strong relatedness structures.

---

> > > > ### Public Comment · ~Ahmad_Beirami1 · 2020-11-13
> > > > **The issue in your argument**
> > > >
> > > > Let's make the problem simpler. Let ${\cal P} = ${$P_1, P_2$}. Let $P_1$ be a Bernoulli source with probability of observing 1 to be $0.5-\epsilon$ and let $P_2$ be a Bernoulli source with probability of observing 1 to be $0.5 + \epsilon$.
> > > >
> > > > Let's also assume that $M=1,$ and $n \to \infty.$ So under mild regularity assumptions we know exactly what the first task is. Let also $k = 1$. Then, the minimax estimator (with no side information) is going to be simple (assuming a well-behaving loss function): $\hat{\theta} = P_1$ if $z = 0$ and $\hat{\theta} = P_2$ if $z=1.$ Now, I tell you that the first task's parameter is indeed $P_1.$ Can you please construct an estimator that beats this? For arbitrary $k$, the minimax estimator is going to be similarly defined to be $P_1$ if $z^k$ has more $0$'s than $1$'s and otherwise; and the fact that Task 1 is either $P_1$ or $P_2$ cannot help improve over it.
> > > >
> > > > The main issue with your argument lies in that you are assuming that $\frac{1}{k}\sum_{z' \in S_{M+1}} z'$ is a minimax estimator in this setup. That is certainly not the case, especially with ${\cal P} = ${$P_1, P_2$}, a finite set. In fact, $\frac{1}{k}\sum_{z' \in S_{M+1}} z'$ is a bad estimator because a priori we know that the source is either Bernoulli($0.5 - \epsilon$) or Bernoulli($0.5+\epsilon$), and this information is not used in your estimator.  I agree that the data dependent estimator $\hat{\theta} = \frac{1}{Mn + k} (\sum_{z \in S_{1:M}} z + \sum_{z' \in S_{M+1}} z')$ does improve over it, but the minimax risk (risk associated with the minimax estimator) remains unchanged. As a side note, minimax estimator depends on the loss function that you are optimizing and might be extremely hard to construct. In general, ML estimator is not minimax optimal.

---

> > > > > ### Comment · Area_Chair1 · 2020-11-15
> > > > > **Pls follow up**
> > > > >
> > > > > Dear Authors,
> > > > >
> > > > > can you pls follow up on this discussion thread ?
> > > > >
> > > > > Thank you,
> > > > >
> > > > > Area Chair

---

> > > > > ### Author Response · Authors · 2020-11-15
> > > > > **We acknowledge issues with our example. Looking back to definition...**
> > > > >
> > > > > We apologize for the delay in our reply.
> > > > >
> > > > > We acknowledge that our comparison to the MLE estimator is misleading and does not obviously provide the guarantee that we asserted --- we apologize for this.
> > > > >
> > > > > It seems difficult for us to reach consensus on this. If we understand correctly, you are not claiming that any of our proofs are incorrect --- only that you disagree with the validity of the minimax risk formulation in the meta-learning setting.
> > > > >
> > > > > ### Addressing definition
> > > > >
> > > > > We disagree with your claim that our definition is meaningless in this setting. However, we would like to point out that several of your suggestions can be incorporated directly into our definition without violating any of our theoretical results.
> > > > >
> > > > > First, we could modify the supremum to require $P_{M+1} \notin \\{P_1,\ldots,P_M\\}$. This requires no changes to any of our proofs. Alternatively, we could require directly that $P_{M+1}$ is always chosen to be within $\alpha$ KL-divergence of $P_1,\ldots,P_M$ in the supremum. We would need to add this KL constraint to Theorem 1 (making it slightly less general) but this is not problematic as we introduce this constraint for all later results anyway.
> > > > >
> > > > > Second, we can freely modify the infimum to search over a restricted space of $\hat{\theta}$. We make no assumptions about the form of the estimator in our proof of Theorem 1. (Though this approach is likely to introduce looseness in the bounds).
> > > > >
> > > > > ### Addressing original claim
> > > > >
> > > > > Looking back to your original proof, there are a couple points that we find problematic and would appreciate your thoughts on.
> > > > >
> > > > > > "Now, $R$ is constant with respect to $P_1,\ldots,P_M$ ($\hat{\theta}$ would not depend on $P_1,\ldots,P_M$ which are independent of $P_{M+1}$"
> > > > >
> > > > > This point is not obvious to us. The estimator may certainly depend on $P_1,\ldots,P_M$, as it is a two-stage learner that first processes data from these tasks. Could you clarify this?
> > > > >
> > > > > > "If regularity conditions hold for the minimax risk to satisfy a minimax theorem"
> > > > >
> > > > > We have no reason to expect that such regularity conditions hold. Indeed, as you have pointed out, the RHS is equal to the standard minimax risk for which we do not expect to have a strong minimax property like this.

---

> > > > > > ### Public Comment · ~Ahmad_Beirami1 · 2020-11-15
> > > > > > **Thanks for the clarifications. More on the problem setup.**
> > > > > >
> > > > > > Thank you for the detailed response.
> > > > > >
> > > > > > **It seems difficult for us to reach consensus on this. If we understand correctly, you are not claiming that any of our proofs are incorrect --- only that you disagree with the validity of the minimax risk formulation in the meta-learning setting.**
> > > > > >
> > > > > > I have not had a chance to follow the proofs; my main comments so far are on the problem setup and the form of the claimed results.
> > > > > >
> > > > > > **First, we could modify the supremum to require $P_{M+1} \notin \{P_1,\ldots,P_M\}$. This requires no changes to any of our proofs. Alternatively, we could require directly that $P_{M+1}$ is always chosen to be within $\alpha$ KL-divergence of
> > > > > > $P_1,\ldots,P_M$ in the supremum. We would need to add this KL constraint to Theorem 1 (making it slightly less general) but this is not problematic as we introduce this constraint for all later results anyway.**
> > > > > >
> > > > > > $P_{M+1} \notin \{P_1,\ldots,P_M\}$ may only help lower the novel task's minimax risk if $\cal P$ is a finite set. It will not resolve the problem for an uncountable $\cal P$. It is unclear to me how to characterize the improvement even in the finite $\cal P$ case for finite $n$. If we revisit the toy example with two tasks, and assume that the first task and the novel task are different, can you please characterize the minimax risk with side information in this case and show that it is smaller than that with no side information?
> > > > > >
> > > > > > BTW, the entire Section 4 is built on the assumption that $\cal P$ is a finite set. However, that assumption on its own is unrealistic for meta-learning. Why would we suspect that the task of interest is only from a finite set of possible tasks, and then we also assume that the novel task is different from the other tasks ($P_{M+1} \notin \{P_1,\ldots,P_M\}$)? To me, this is a very non-standard problem setup for meta-learning. This sounds more like a guessing problem where one would like to guess the task ID, and you will be ruling out some possibilities for the task ID with side information. Can you please comment?
> > > > > >
> > > > > > **Second, we can freely modify the infimum to search over a restricted space of $\hat{\theta}$. We make no assumptions about the form of the estimator in our proof of Theorem 1. (Though this approach is likely to introduce looseness in the bounds).**
> > > > > >
> > > > > > Restricting the set over which you search for $\hat{\theta}$ can only hurt as you also mention in parentheses. I am not sure why that is a good idea and what problem it is supposed to address.
> > > > > >
> > > > > >
> > > > > > Regarding the proof sketch (I did not mean this to be perceived as a rigorous proof) that I provided in the first comment.
> > > > > >
> > > > > > **This point is not obvious to us. The estimator may certainly depend on $P_1,\ldots,P_M$, as it is a two-stage learner that first processes data from these tasks. Could you clarify this?**
> > > > > >
> > > > > > In this case, $P_{M+1}$ is statistically independent of $P_1,\ldots,P_M$ by construction and it is also independent of any samples from them. Hence, an optimal estimator cannot depend on $P_1,\ldots,P_M$.
> > > > > >
> > > > > > **We have no reason to expect that such regularity conditions hold. Indeed, as you have pointed out, the RHS is equal to the standard minimax risk for which we do not expect to have a strong minimax property like this.**
> > > > > >
> > > > > > With $\cal P$ a finite set, which is the case in your paper, assuming some compactness and convexity on the loss function such regularity conditions should be verifiable, which is the case for the toy example that we discussed earlier. If you make explicit a problem setup with an explicit loss function, and explicit $\cal P$, I'd be happy to provide more explicit conditions. Please see Merhav and Feder for a version of such minimax results.
> > > > > >
> > > > > > Merhav, N. and Feder, M., 1995. A strong version of the redundancy-capacity theorem of universal coding. IEEE Transactions on Information Theory, 41(3), pp.714-722.

---

> > > > > > > ### Author Response · Authors · 2020-11-16
> > > > > > > **Thanks for the response. Some clarifications and problems**
> > > > > > >
> > > > > > > "$P_{M+1} \notin \\{P_1,\ldots,P_M\\}$ may only help lower the novel task's minimax risk if $\mathcal{P}$ is a finite set."
> > > > > > >
> > > > > > > We certainly constructed this condition with finite sets in mind. However, we could also include a continuous separation condition (i.e. delta-separation introduced later in Section 4) for uncountable $\mathcal{P}$ (a condition that could already apply in Theorem 1).
> > > > > > >
> > > > > > > "can you please characterize the minimax risk with side information in this case and show that it is smaller than that with no side information?"
> > > > > > >
> > > > > > > In your toy example, if we know the identity of the first distribution, we know by exclusion the identity of the novel task distribution (as there are only 2). Even with finite $n$, we increase the probability that we correctly identify the novel task distribution with $k=1$.
> > > > > > >
> > > > > > > We also included a close-in-KL condition that you do not discuss, which does not have the issues you raised with the exclusion condition.
> > > > > > >
> > > > > > > In any case, we raised these points to highlight to yourself (and other readers), that even in the case that you are correct about our definition in Section 3, our theoretical results can adapt without additional work to cover your suggested changes.
> > > > > > >
> > > > > > > "BTW, the entire Section 4 is built on the assumption that $\mathcal{P}$ is a finite set"
> > > > > > >
> > > > > > > This is absolutely not true. Our bounds in Section 4 apply to uncountable $\mathcal{P}$. For example, we apply them to a space of linear regression models in Section 5. The finite sets that we introduce in our results are local-packing sets, a common tool in minimax lower bounds.
> > > > > > >
> > > > > > > Noting this, we maintain that it is not possible in general to provide regularity conditions necessary for "maxmin = minmax". One could look to existing minimax literature for an example of such problems, e.g. Raskutti et al. 2011.
> > > > > > >
> > > > > > > We are glad to have the opportunity to discuss these issues with you, and are thankful that this discussion has already identified several parts of our paper that when examined carefully from the outside could definitely use further clarification. However, we worry that you may be seeking out errors in our paper without a full appreciation of all of its technical content. We are by no means claiming that our paper is without fault. As with any scientific publication under review (and beyond...), mistakes are possible and we are open to identifying and discussing them with you. We ask that we both take a step back and continue this discourse at a steadier pace, after reviewing each others' references and comments.

---

### Public Comment · ~Ahmad_Beirami1 · 2020-11-14
**Corollary 1, Corollary 2, Theorem 2**

In Corollary 1, you claim a lower bound of the form
$R_{\cal P}^\star \geq \psi(\delta) \left(1 - \frac{ 1+ \left(\frac{Mn}{J-1}+k\right)\alpha}{\log J}\right).$
As $n \to \infty$, this bound is vacuous (the right hand side becomes negative and even diverges to $-\infty$), which is the regime of interest in this paper, i.e., having $k$ samples from a novel task and $n$ samples from related tasks where $n \gg k$.


In Corollary 2, assuming that $M+1<J$, you claim a bound of the form
$R_{\cal P}^\star \geq \psi(\delta) \left( \frac{\log (J-M) - k\alpha - 1}{\log J}\right).$
This bound has no dependence on $n$. For example, setting $n=0,$ it is clear that $R^\star_{\cal P} = R^\star$ and this is a looser bound on $R^\star.$ In fact, one can choose $M = J-2$ and make the bound the loosest.


In Theorem 2, you claim a bound on the minimax risk of the form
$R^\star \geq \psi(\delta) \left(1 - \frac{ 1+ k\alpha}{\log J}\right).$
This bound is vacuous and very loose as $k\to \infty,$ as the right hand side becomes negative for sufficiently large $k$ and potentially diverges to $-\infty$. This bound clearly does not have the right scaling with $k,$ which is a basic desirable property.

---

> ### Comment · AnonReviewer2 · 2020-11-14
> **Some of my ideas on information-theoretical bounds**
>
> Hi Ahmad,
>
> I would like to say something positive for this paper.
>
> First of all, notice that $\psi(\delta)$ can introduce some scaling factor, e.g. under the Corollary 1 the authors claimed that the bound requires $\psi(\delta) = \frac{1}{k}$ or something similar, which make the bound not necessarily diverge.
>
> Secondly, I would like to say that, most of the information-theoretical bound are not designed for considering the risk when we have ''infinite'' information, as you mentioned. You may notice that even when we  proper scale with $\psi(\delta)$, the bound can still be trivial in the asymptotic. But in my opinion, what we really care is about the limit we can achieve within the ''finite'' information, from the source task and the target task. And I feel the authors really do something on that, though not perfect. In my review I say that the KL may not be the proper measure to measure the task similarity, and thus can be pessimistic. But given that little work have been done on the information-theoretical consideration of meta-learning, I still give a positive score. If you know some similar work, I would be happy to have a read and update my score accordingly. But now I still think this paper can introduce some new perspectives on meta-learning.
>
> Thanks.

---

> > ### Public Comment · ~Ahmad_Beirami1 · 2020-11-15
> > **Thanks for sharing your thoughts**
> >
> > **I would like to say something positive for this paper.**
> >
> > Yes, there are positive aspects of this paper as well. I really enjoyed the example in Section 5. The issue is that it is disconnected from the setup in Section 3 and the theoretical study in Section 4 (Those two sections are also disconnected from each other as I mentioned in the comments below -- The setup in Section 3 does not support the results in Section 4).  It would have been nice if the authors built some theory that was applicable to the setup of Section 5 rather than claiming general bounds.
> >
> >
> > **First of all, notice that $\psi(\delta)$ can introduce some scaling factor, e.g. under the Corollary 1 the authors claimed that the bound requires $\psi(\delta) = \frac{1}{k}$ or something similar, which make the bound not necessarily diverge.**
> >
> > In Corollary 1, for any $n \geq \frac{(J-1)\log J}{M}$ the right hand side evaluates to a negative value. In fact, fixing all other parameters (including $k$) and letting $n \to \infty,$ the right hand side diverges to $-\infty$. I am not sure how $\psi(\delta)$ can play a role to save the day.
> >
> >
> > **Secondly,  I would like to say that, most of the information-theoretical bound are not designed for considering the risk when we have infinite information, as you mentioned. **
> >
> > I disagree with you. Information theory is the discipline of asymptotic analysis. Starting from Shannon's "THE Mathematical Theory of Communication" paper and the definition of channel capacity (http://people.math.harvard.edu/~ctm/home/text/others/shannon/entropy/entropy.pdf), information theory has strived to find lower bounds on different problems, such as compression, security, communication, that are asymptotically tight by constructing matching upper bounds through some achievability analysis.
> >
> >
> > **You may notice that even when we proper scale with $\psi(\delta)$, the bound can still be trivial in the asymptotic. But in my opinion, what we really care is about the limit we can achieve within the ''finite'' information, from the source task and the target task. And I feel the authors really do something on that, though not perfect.**
> >
> > I agree with you that non-asymptotic bounds are preferable, especially in the real world and that is where the information theory has been less developed although there are many works on fine-length or finite block length information theory in the past decade. One basic property of useful non-asymptotic bounds is that their asymptotic behavior should match the asymptotic expected behavior. Then, one starts analyzing the tightness of such bounds. Unfortunately, the bounds in this paper have the wrong scaling with $n$, $k$, $d$ and there is no hope for them to be tight.
> >
> >
> > ** But given that little work have been done on the information-theoretical consideration of meta-learning, I still give a positive score. If you know some similar work, I would be happy to have a read and update my score accordingly. But now I still think this paper can introduce some new perspectives on meta-learning.**
> >
> > The problem of source compression is a simple instance of a learning problem with KL divergence loss, and there is a rich body of literature on that. In fact, the counter-example that the authors construct is covered by this rich body of literature. I just cite a few relevant papers here that construct minimax estimators and/or develop upper/lower bounds on minimax risk in the universal source coding problem; there is also a large body of literature on source coding with side information that I am not even citing here.
> >
> > - Clarke, B.S. and Barron, A.R., 1994. Jeffreys' prior is asymptotically least favorable under entropy risk. Journal of Statistical planning and Inference, 41(1), pp.37-60.
> >
> > - Merhav, N. and Feder, M., 1995. A strong version of the redundancy-capacity theorem of universal coding. IEEE Transactions on Information Theory, 41(3), pp.714-722.
> >
> > - Xie, Q. and Barron, A.R., 1997. Minimax redundancy for the class of memoryless sources. IEEE Transactions on Information Theory, 43(2), pp.646-657.
> >
> > - Barron, A., Rissanen, J. and Yu, B., 1998. The minimum description length principle in coding and modeling. IEEE Transactions on Information Theory, 44(6), pp.2743-2760.
> >
> > While I am less familiar with the literature on the theory of meta-learning, here are some relevant existing studies:
> >
> > - Mehta, N., Lee, D. and Gray, A.G., 2012. Minimax multi-task learning and a generalized loss-compositional paradigm for MTL. In Advances in Neural Information Processing Systems (pp. 2150-2158).
> >
> > - Amit, R. and Meir, R., 2018, July. Meta-learning by adjusting priors based on extended PAC-Bayes theory. In International Conference on Machine Learning (pp. 205-214).
> >
> > - Mousavi Kalan, M., Fabian, Z., Avestimehr, S. and Soltanolkotabi, M., 2020. Minimax Lower Bounds for Transfer Learning with Linear and One-hidden Layer Neural Networks. Advances in Neural Information Processing Systems, 33.

---

> > > ### Comment · AnonReviewer2 · 2020-11-15
> > > **Thanks for your reply. I mostly agree with your thoughts.**
> > >
> > > Probably I may not express my idea on information-theoretical bound clearly. But definitely we would like to find some lower bounds from the information-theoretical analysis. I guess your main argument is that, these lower bounds are not asymptotically reasonable. And what I want to say is the authors give some results on the small $n$, $k$ regime. I would also like to hear from the authors on their thoughts of the asymptotic behavior.
> > >
> > > Also thanks for pointing out these relevant papers. I may adjust my review based on the references you mentioned, especially the minimax risk part.

---

> > > ### Author Response · Authors · 2020-11-15
> > > **Tightness of these results**
> > >
> > > Hi Ahmad,
> > >
> > > We appreciate your continued interest in our work!
> > >
> > > We would first point out that Theorem 2 is not a novel result in this work, and is a standard bound from existing minimax literature (as cited in our work).
> > >
> > > In terms of the purported divergence of these bounds, along the lines of Reviewer 2's comment, note that $\alpha$ is also typically chosen to depend on $n$, $\delta$, and other problem-dependent parameters. See our proof of Theorem 3, for example.
> > >
> > > Further, our Corollary 2 loses dependence on $n$ via the data-processing inequality. Therefore, we would expect these bounds to be tightest when $n \rightarrow \infty$, not $n=0$.
> > >
> > > Thank you for your references too, in particular we were (somehow) unaware of Kalan et al. 2012, which certainly looks related. We will read this paper more deeply, but note that their minimax risk does not consider a novel-task generalization but rather average performance over a set of tasks.

---

> > > > ### Public Comment · ~Ahmad_Beirami1 · 2020-11-15
> > > > **More on tightness of these results.**
> > > >
> > > > **We would first point out that Theorem 2 is not a novel result in this work, and is a standard bound from existing minimax literature (as cited in our work).**
> > > >
> > > > Unfortunately, Theorem 2 is not a standard minimax result, and is certainly a problematic result on its own right because it lacks the proper scaling with $k$ (unless there are certain dependences of $\alpha$ and $\psi$ on other parameters that I am failing to see). Please see the cited references in my previous comment for standard minimax results in the literature from 90s (which apply to the toy counter example discussed in the other thread). Does the existing cited work discuss the scaling of this bound with $k$? Does it discuss its tightness?
> > > >
> > > > **In terms of the purported divergence of these bounds, along the lines of Reviewer 2's comment, note that  is also typically chosen to depend on $n$, $\delta$, and other problem-dependent parameters. See our proof of Theorem 3, for example.**
> > > >
> > > > First, please make all dependences on all parameters explicit in the paper.
> > > > Second, can you please refute that "the bounds in Corollary 1 becomes negative for large enough $n$"?
> > > >
> > > > **Further, our Corollary 2 loses dependence on  via the data-processing inequality. Therefore, we would expect these bounds to be tightest when $n \rightarrow \infty$, not $n=0$.**
> > > >
> > > > Well, you claimed Corollary 2 as a **tighter** bound in the paper. Can you please clarify in what sense it is a tighter bound then?
> > > >
> > > > Finally, can you please chime in on the asymptotic scaling of these bounds with different parameters?
> > > >
> > > > Can you at least evaluate these results on the toy counter-example from the other thread and show that they are meaningful, and scale correctly?

---

> > > > > ### Public Comment · ~Ahmad_Beirami1 · 2020-11-16
> > > > > **Theorem 2 in this paper is not Theorem 1 of (Loh 2017)**
> > > > >
> > > > > I just checked Loh (2017). The bound in their Theorem 1 is of the form
> > > > > $R^\star \geq \psi(\delta) \left(1 - \frac{ I(Z^n; P_{M+1}) - 1}{\log J}\right).$
> > > > >
> > > > > Their bound's scaling is dramatically different from your Theorem 2. In particular, their right hand side is always non-negative because $I(Z^n; P_{M+1}) \leq \log J$, whereas yours diverges to $-\infty$ as $k \to \infty.$
> > > > >
> > > > > Can you please point us to the literature where Theorem 2 is stated in the form claimed in your paper?

---

> > > > > > ### Author Response · Authors · 2020-11-16
> > > > > > **With a few very small steps, Theorem 1 (Loh 2017) => Theorem 2 (ours)**
> > > > > >
> > > > > > We apologize if the presentation of Theorem 2 is misleading. As this has raised significant confusion, we will work on improving the presentation in this section to make things clearer.
> > > > > >
> > > > > > We adopted the approach of Khas'minskii, 1979 and collapsed the mutual information under iid samples for these corollaries. This allows us to reason about the sample size within the bound, which was the central object of interest for us in Section 4. This loosening is discussed additionally in Yang & Barron, 1999.
> > > > > >
> > > > > > Looking to Loh 2017, our bound can be recovered by substituting the results of Loh's Lemma 2 into their Theorem 1, and using the above loosening. The bound that you provided does not include this local-packing, which we clearly state is utilized in our work. Your stated bound is more comparable to our Theorem 1, which is written in terms of mutual information alone.
> > > > > >
> > > > > > We could keep the bounds in the corollaries in terms of KL-divergence (as in Loh), which are more familiar, but we wanted to be explicit in sample size to facilitate discussion in Section 3. This is not necessary, but we felt provided valuable insight into the comparison of the meta-learning minimax bounds with their iid counterparts.
> > > > > >
> > > > > > In our minimax bounds, $\delta$ and $\alpha$ need to be set according to the problem space and loss function of interest. See Raskutti et al, 2011, Section III, A.3 for some related discussion. We explore only one technique (local-packing), but Theorem 1 is general and other techniques could be applied. We will make these dependencies clear in a revised version.
> > > > > >
> > > > > > > "can you please refute that "the bounds in Corollary 1 becomes negative for large enough $n$?"
> > > > > >
> > > > > > See Theorem 3, for example, which uses Corollary 1 to derive a bound which does not become negative for large $n$.
> > > > > >
> > > > > > > "Can you please clarify in what sense it is a tighter bound then?"
> > > > > >
> > > > > > In the setting where we know that the environment is only partially observed $M + 1< J$, we do not expect the error to shrink to zero as $n$ grows. In this sense, Theorem 1 is only a loose lower-bound. Corollary~2 formalizes this intuition, and provides a lower-bound that does not shrink to zero with $n \rightarrow \infty$. This is what we mean by a tighter bound. We agree that this can be clarified in the paper.
> > > > > >
> > > > > > > Finally, can you please chime in on the asymptotic scaling of these bounds with different parameters?
> > > > > >
> > > > > > This is difficult to summarize in general, particularly in this forum. We discuss this in our paper. Bounding the mutual information is highly problem-specific and naturally different asymptotic behaviours arise.
> > > > > >
> > > > > > > Can you at least evaluate these results on the toy counter-example from the other thread and show that they are meaningful, and scale correctly?
> > > > > >
> > > > > > There are a lot spinning plates in this rapidly growing discussion. To get this response to you in a timely manner, we will not provide an analysis of this toy problem at the moment. In any case, we do not see how this would be generally convincing or otherwise necessary.
> > > > > >
> > > > > > References
> > > > > >
> > > > > > Loh, 2017, "On Lower Bounds for Statistical Learning Theory"
> > > > > >
> > > > > > Rafail Z Khas’minskii, 1979, "A LOWER BOUND ON THE RISKS OF NON-PARAMETRIC ESTIMATES OF DENSITIES IN THE UNIFORM METRIC"
> > > > > >
> > > > > > Yang & Barron, 1999, "Information-theoretic determination of minimax rates of convergence"
> > > > > >
> > > > > > Raskutti et al., 2011 "Minimax rates of estimation for high-dimensional linear regression over $\ell_q -balls$"

---

> > > > > > > ### Author Response · Authors · 2020-11-16
> > > > > > > **A useful pointer: Our Appendix B through B.1**
> > > > > > >
> > > > > > > We should have included in our previous response that the derivation we describe above is shown in full in our appendix, Sections B and B.1.

---

> > > > > > > ### Public Comment · ~Ahmad_Beirami1 · 2020-11-16
> > > > > > > **The bound in Theorem 3 is vacuous as $n \to \infty$.**
> > > > > > >
> > > > > > > **See Theorem 3, for example, which uses Corollary 1 to derive a bound which does not become negative for large $n$.**
> > > > > > >
> > > > > > > I specifically checked Theorem 3 per your request. In the proof, there is an unnumbered equation on Page 16 where you claim a bound of the form
> > > > > > > $R^\star_{{\cal P}_{LR}} \geq \delta^2\left( 1 - \frac{(nM2^{-d} + k) 32 \delta^2/\sigma^2 + 1}{d}\right)$
> > > > > > >
> > > > > > > Can you please explain how this bound will not become negative for large $n$?

---

> > > > > > > > ### Author Response · Authors · 2020-11-16
> > > > > > > > **This follows from the next 2 lines**
> > > > > > > >
> > > > > > > > > "Can you please explain how this bound will not become negative for large $n$?"
> > > > > > > >
> > > > > > > > In the two lines of the proof following this equation, we choose a value for $\delta^2$ which guarantees that this term is positive (in fact: $\geq 1/4)$).
> > > > > > > >
> > > > > > > > (In fact, there was a constant error in the proof that we just spotted. This was trivial to fix and is corrected in the latest version.)

---

> > > > ### Comment · AnonReviewer2 · 2020-11-15
> > > > **Can the authors make a revision based on Ahmad's suggestions?**
> > > >
> > > > In terms of the discussions here, I suggest the authors to make a revision based on the issues raised by Ahmad, e.g. make the dependence on parameters explicit, so that we can understand and evaluate the contribution of this paper.
> > > >
> > > > Thanks!

---

> > > > > ### Author Response · Authors · 2020-11-16
> > > > > **We agree that this is valuable**
> > > > >
> > > > > Though we replied to you directly, we want to state here too that we will be including several changes in our revised version based on this discussion (and the other thread).
> > > > >
> > > > > We plan to include discussion of the references that Ahmad provided that we are missing. We will also clearly described the role of $\alpha$ and $\delta$, and generally improve clarity around Theorem 2 and our corollaries in Section 4 (whose details are currently too hidden in the appendix).

---

> ### Comment · AnonReviewer4 · 2020-11-21
> **The bounds seem useless as pointed out by Ahmad**
>
> Ahmad has a very valid point. The bounds are useless. He is right that the implicit regime of interest here is $n \texttt{>>} k$ and $n \to \infty$. Not only from an information theoretic point of view but also from an algorithmic theory point of view we can argue that this is the regime of interest. Any bound that diverges to $-\infty$ in this regime is even more useless than a simple lower bound of 0. I don't think that this paper is worthy of publishing unless the authors could provide bounds that work for these regimes. So I am inclined to change my assessment from accept to reject unless there is a fix for these concerns.

---

> > ### Author Response · Authors · 2020-11-21
> > **The bounds are not useless: the packing set construction can prevent negative bounds**
> >
> > Thank you for your comment.
> >
> > This issue stems from a misinterpretation of these bounds. This bound has the right kind of behavior, matching the intuition that the minimax risk should decrease with increasing n: seeing more examples of the $M$ tasks can decrease the error on the novel task. The asymptotic behavior, as n goes to infinity is not an issue. The packing set can be constructed with knowledge of the sample size --- therefore both $\delta$ and $\alpha$ may depend on $n$, $M$, and $k$. This prevents the bound from becoming negative (as is the case in our proof of Theorem 3). We have clarified this in our latest revision.
> >
> > In fact, one of the references provided by Ahmad has a bound of this form too: "Minimax Lower Bounds for Transfer Learning with Linear and One-hidden Layer Neural Networks", Kalan et al. Equations 5.2 and 5.3 in their appendix also _appear_ to have negative risk for large $n$. This is not the case.

---

### Public Comment · ~Ahmad_Beirami1 · 2020-11-16
**Major problem with Theorem 3**

In Theorem 3, the authors claim a bound of the form
$R^\star_{{\cal P}_{LR}} \geq O\left(\frac{d\sigma^2}{\gamma^2 (2^{-d} nM +k)} \right)$.

Nothing in the Theorem statement hints at the fact that the authors are assuming $2\delta$-separability. However, it turns out that in the proof attempt, the authors invoke Corollary 1 with a particular form of $\delta$ given by
$\delta^2 = d\sigma^2/64 \gamma^2(2^{-d} nM +k).$

More troubling is the fact that $\delta$ which is a property of concept class suddenly depends on $d$, $\sigma$, $\gamma$, $n$, $M$, and $k$ in an unexplained way.

Even more troubling is that $\delta \geq 1$ for large enough $n$ or $k$. Hence, the concept class ${\cal P}_{LR}$ becomes empty for large $n$ or $k$ because the authors are assuming that the parameters live in $B_2(1)$, and hence at most one parameter could exist in the unit ball with $2$-separability.

**I am stopping to post more public comments under this submission. If the authors or the reviewers would like to hear from me, I can be reached via email.**

Cheers!
Ahmad

---

> ### Author Response · Authors · 2020-11-16
> **There are several significant misunderstandings here**
>
> First, we sincerely thank you for your interest in our submission and the discussion that you have generated. We intend to introduce several clarifications to our paper as a direct result of these interactions.
>
> > "Nothing in the Theorem statement hints at the fact that the authors are assuming $2\delta$-separability"
>
> We do not need to assume separability in the Theorem statement. Within the proof we construct a local-packing set with bounded cardinality for the problem space. This has the desired $\delta$ separation, so that we can invoke Corollary 1.
>
> > "Even more troubling is that $\delta \geq 1$ for large enough $n$ or $k$"
>
> In fact, $\delta$ decreases with $n$, $M$, and $k$. This should be read as, $\delta^2 = \frac{d\sigma^2}{64\gamma^2(2^{-d}nM + k)}$. This is clear from how we use $\delta^2$ after it's value is assigned. However, we will improve the equation formatting to make this clearer in the revised version. You do raise a good point that we should include necessary conditions on the problem dimensions for this packing set to be well-defined --- we have included these explicitly now too.
>
> In general, you seem to have misunderstood this theorem and proof. The first step in our proof is to find a packing set of $\mathcal{P}_{LR}$ which satisfies $\delta$-separability and the KL-divergence constraint. To do this, we use a (admittedly confusing but common) scaling trick, that we also discuss in our response to Reviewer #1. This allows us to embed the packing set within the unit ball, with any $\delta < 1/2$ --- that is, we have built an uncountable set of valid local-packings. We then invoke our lower bound results using these packing sets, and choose a $\delta$ depending on the problem parameters to retrieve the desired bound.
>
> Several related proofs can be found in Raskutti et al., 2011. In particular, our proof follows a similar format to the proof of their Theorem 1(b).
>
> If you remain interested, we would be very happy to continue our discussion offline once our anonymity is lifted.

---

### Public Comment · ~Dror_Baron2 · 2020-11-16
**Simply weird**

I tried to read this paper. Notations and terms are often undefined, and if so in a haphazard way. As other commenters have mentioned, the results are questionable. Did you even prove your corollaries? For example, on page 4 you wrote that Corollary 1 was derived from Theorem 1. Where is the derivation?
As I wrote in the title of my comment, this paper is simply weird.

---

> ### Author Response · Authors · 2020-11-16
> **An embarrassing omission**
>
> You are unfortunately correct that the proof of Corollary 1 is not stated anywhere in the paper... We missed this as (fortunately) the proof is a one-liner given the results proved in the appendix. The proof follows directly from Theorem 1, Lemma 3, and Lemma 4. We have, of course, added this to the appendix.
>
> Proofs for all other results are given in the appendix (we double checked).
>
> Regarding your other comments, could you please be specific regarding the notation that you found confusing? This would help us to improve clarity.

---

### Author Response · Authors · 2020-11-16
**Overview of current discussions**

We have received a high volume of comments on our paper. Some of these are valid points that we are addressing in the current version. However, it should be noted that these involve clarifications, notational issues, further discussion of relevant appendix details in the main paper, and **not** errors in the paper. Following this lengthy discussion, we provide a summary of the broad points here.

### Claim: Section 4 considers only finite $\mathcal{P}$

This is not true. We consider uncountable $\mathcal{P}$, and produce our bounds via finite local packings of the space --- a standard technique discussed in e.g. Loh, 2017.

### Claim: Theorem 3 (and its proof) is invalid

This is incorrect, and stems from a misunderstanding of the proof. First, it is claimed that $\delta$ must be a property of the concept class, but our bounds apply with any valid packing set produced for $\mathcal{P}$. It is standard practice to introduce dependence on the sample size and other problem parameters in these results (see e.g. Raskutti et al. 2011, proof of Theorem 1.b).

Second, it is claimed that $\delta \geq 1$ for large $n$. This is also false ($\delta$ decreases) and is easily verifiable from the proof. We have made this absolutely clear in the proof.

### Corollary 1, Corollary 2, Theorem 2

Ahmad pointed out that fixing all other parameters and taking $n\rightarrow\infty$ leads to a negative bound. However, the parameters $\delta$ and $\alpha$ can depend on $n$ itself --- see Theorem 3 in our work, or similar approaches in Raskutti et al. 2011 for examples. We will point out that these can depend on sample size and properties of $\mathcal{P}$ in the main paper.

Regarding Corollary 2, Ahmad stated that this bound is loose when $n=0$. We agree with this assessment, but this is besides the point. As stated in our paper, Corollary 2 is derived via application of the data processing inequality. As such, we expect that it will be tightest when $n \rightarrow \infty$ and there is infinite data from each of the meta-training tasks available. We will clarify this further in our paper.

Further, Ahmad argued that Theorem 2 in our paper is invalid. We pointed out that this can be derived directly from existing results in the literature (which was fully detailed in our appendix at the time of submission). We will summarize this derivation in the main paper.

### Claim: $R^* = R_{\mathcal{P}}^*$

Ahmad claims that the side information has no effect on our proposed minimax risk. We understand that the current notation allows for this interpretation and are working on clarifications to the definition and theorem statement to clarify the differences between the minimax risk with and without side information ($R^*$ and $R_{\mathcal{P}}^*$). We are confident that these changes may be included without significant modification of our results. We are putting together a concrete example to further clarify this relationship and will include this alongside any necessary modifications to Theorem 1 in the next few days.

### References

Loh, 2017, "On Lower Bounds for Statistical Learning Theory"

Rafail Z Khas’ minskii, 1979, "A lower bound on the risks of non-parametric estimates of densities in the uniform metric"

Raskutti et al., 2011 "Minimax rates of estimation for high-dimensional linear regression over $\ell_q -balls$"

---

### Author Response · Authors · 2020-11-21
**Uploaded latest revision**

### Summary of changes

In addition to several improvements to clarity following reviewer feedback, we have made a modification to our minimax definition that requires minimal changes to the presented theoretical results. We now require the novel task to be within $\beta$ KL-divergence of the meta-training tasks in the definition of $R_{\mathcal{P}}^*$. This provides a clear differentiation between the iid and meta-learning minimax risks (see example below). The necessary changes to the rest of the paper are summarized below:

- In the statement of Theorem 1, we additionally require the $\beta$ KL constraint in the packing set. No change to proof needed.
- No changes needed to corollaries (though we clarified presentation)
- In Theorem 3, we require an additional condition depending on $\beta$

In our bounds, $\beta$ effectively restricts our choice of packing set to those that satisfy the required KL constraint. This affects the proof of Theorem 3, where we construct such a packing. Some informal intuition about the packing set may be useful: we can visualize it as building a
packing of a sphere. Decreasing delta reduces the radius of the
sphere and packs the points more closely. The beta constraint
means that we are only allowed to consider those packings where
the sphere has radius at most beta.

### An example where $R^* \neq R_{\mathcal{P}}^*$

Consider 4 distributions,

$\mathcal{P} = \\{ Ber(0.2), Ber(0.3), Ber(0.7), Ber(0.8) \\},$

where we observe 1 sample, $x$, from one of these distributions. In the IID setting, the best we can do is to guess $Ber(0.8)$ if we see heads and $Ber(0.2)$ if we see tails. Looking at the conditional probabilities:

$P(p=0.2 | x = 1) = 0.1$

$P(p=0.3 | x = 1) = 0.15$

$P(p=0.7 | x = 1) = 0.35$

$P(p=0.8 | x = 1) = 0.4$

So we are wrong 60\% of the time (the other case is symmetrical).

On the other hand, in the meta learning setting if we choose $\beta = 0.1$ with $M = 1$, then the possible configurations of tasks are as follows:

Meta-training $p = 0.2$ or $0.3$; Novel $p = 0.2$ or $0.3$

Meta-training $p = 0.7$ or $0.8$; Novel $p = 0.7$ or $0.8$

For a total of 8 possible settings. [The KL between the closer distributions is $\approx 0.026$, and $\approx 0.339$ for $Ber(0.3)$ and $Ber(0.7)$ ]

If we observe many samples in the meta-training task (large n), then we know that the novel task can be from only one of two distributions. Hence our error rate with one sample will be at most 50\% (slightly better if we actually compute it).

---

> ### Comment · AnonReviewer1 · 2020-11-23
> **A quick question**
>
> Thanks for your replies to the comments raised by the general public and other reviewers.
> As far as I understand, $n \to \infty$ is not a concern in corollary 1, since $\delta$ can depend on the model parameters like $n, M, k$. And an application was shown in theorem 3.
>
> The other concern that was raised was whether $R^{\ast} = R^{\ast}_{\mathcal{P}}$. And the authors provided one example where this is definitely not the case. My major concern is whether the example satisfies the conditions in Theorem 1.  The concern arises from the fact that the theorem needs the KL-divergences between the different distribution $\mathcal{P}_j$ to be bounded by $\beta$. I am concerned if a large $\beta$ can make the bounds vacuous.

---

> > ### Author Response · Authors · 2020-11-25
> > **Thank you for the question**
> >
> > Thanks for your response.
> >
> > You are correct that as $\beta$ grows larger the relatedness between the tasks drops. Regarding the example, we designed this example to illustrate the differences between $R_{\mathcal{P}}^*$ and $R^*$ (as this issue arose in prior discussion) and not necessarily as an application of our lower bounds.
> >
> > In terms of the lower bounds, we consider Theorem 3 to serve as a more complete example of how these bounds work under different values of $\beta$. Intuitively, $\beta$ acts as a constraint on the meta-training tasks mutual information term in Theorem 1.

---

### Decision · Program_Chairs · 2021-01-07
**Final Decision**

**Decision:**

Accept (Poster)

**Comment:**

The authors study the theoretical performance of a meta-learning in two settings. In the first one the overall number of possible tasks is limited and tasks are close in KL-divergence. The second setting is MAP estimation (in a hierarchical Bayesian framework) for a family of linear regression tasks. Lower and upper bounds are provided on minimax parameter estimation error.
This paper has spurred a lot of discussion among reviewers and (competent) external commentators. Most of these criticisms were right on target, but the authors managed to convince the reviewers and myself that there was simply an issue of presentation of the main results. I suggest the authors to take into serious considerations all the aspects raised by the reviewers that has generated misinterpretations of the presented results.